# Activin A marks a novel progenitor cell population during fracture healing and reveals a therapeutic strategy

Lutian Yao[1,2], Jiawei Lu[3], Leilei Zhong[3], Yulong Wei[3], Tao Gui[3], Luqiang Wang[3], Jaimo Ahn[4], Joel D Boerckel[3], Danielle Rux[1], Christina Mundy[1], Ling Qin[3]*, Maurizio Pacifici[1]*

[1]Translational Research Program in Pediatric Orthopaedics, Division of Orthopaedic Surgery, Children's Hospital of Philadelphia, Philadelphia, United States; [2]Department of Orthopaedics, The First Hospital of China Medical University, Shenyang, China; [3]Department of Orthopaedic Surgery, Perelman School of Medicine, University of Pennsylvania, Philadelphia, United States; [4]Department of Orthopaedic Surgery, Michigan Medicine, University of Michigan, Ann Arbor, United States

**Abstract** Insufficient bone fracture repair represents a major clinical and societal burden and novel strategies are needed to address it. Our data reveal that the transforming growth factor-β superfamily member Activin A became very abundant during mouse and human bone fracture healing but was minimally detectable in intact bones. Single-cell RNA-sequencing revealed that the Activin A-encoding gene *Inhba* was highly expressed in a unique, highly proliferative progenitor cell (PPC) population with a myofibroblast character that quickly emerged after fracture and represented the center of a developmental trajectory bifurcation producing cartilage and bone cells within callus. Systemic administration of neutralizing Activin A antibody inhibited bone healing. In contrast, a single recombinant Activin A implantation at fracture site in young and aged mice boosted: PPC numbers; phosphorylated SMAD2 signaling levels; and bone repair and mechanical properties in endochondral and intramembranous healing models. Activin A directly stimulated myofibroblastic differentiation, chondrogenesis and osteogenesis in periosteal mesenchymal progenitor culture. Our data identify a distinct population of Activin A-expressing PPCs central to fracture healing and establish Activin A as a potential new therapeutic tool.

*For correspondence:
qinling@pennmedicine.upenn.edu (LQ);
pacificim@email.chop.edu (MP)

**Competing interest:** The authors declare that no competing interests exist.

## Editor's evaluation

This important work identified a novel role for Activin A in promoting long bone fracture repair while also demonstrating its therapeutic potential. The evidence supporting the conclusion that Activin A is an important inducer of chondrocyte and osteoblast differentiation that contributes to bone healing is convincing. This work describes novel and valuable findings that will be of interest to both scientists and clinicians in the musculoskeletal field.

## Introduction

Bone is endowed with the ability to heal after fracture and does so effectively in the majority of, but not all, patients (*Einhorn and Gerstenfeld, 2015*; *Loi et al., 2016*). Given its traumatic nature, a bone fracture triggers local hematoma formation followed by an inflammatory cascade that involves innate and adaptive immune responses (*Claes et al., 2012*). These initial steps elicit the expansion and migration of the otherwise quiescent mesenchymal stem and progenitor cells within the periosteum

that over time undergo differentiation into chondrocytes within the body of callus and osteoblasts at its peripheral ends. Subsequent remodeling of callus routinely leads to nearly scar-less bone healing. Thus, fracture healing replicates, and relies on, many of the processes of endochondral and intramembranous ossification through which the skeleton normally develops and grows prenatally and postnatally (*Lefebvre and Bhattaram, 2010*).

Recent studies have aimed to clarify the character of the periosteal progenitors and their roles in the fracture repair process. Building on their earlier reports (*Matthews et al., 2014*; *Matthews et al., 2016*), Matthews et al. characterized a αSMA-expressing slow-cycling, long-term, and self-renewing periosteal progenitor cell population that when ablated, impaired healing (*Matthews et al., 2021*). Another report provided evidence of a periosteum-resident Mx1+αSMA+ stem cell subpopulation that expressed chemokine CCL receptors needed for effective bone healing (*Ortinau et al., 2019*). Despite these and other advances, much remains unclear about the regulation of fracture repair and in particular: what factors set the repair process in motion and in what sequence they act; what distinguishable stages exist along the repair process and how cells move from one stage to the next; how the progenitors commit to separate differentiation lineages; and importantly for patient care, to what extent the progenitor cell's repair capacity can be modified, corrected, or improved. Unbiased single-cell RNA-sequencing (scRNA-seq) approaches have recently been used to characterize mesenchymal cells in marrow, delineating their regulation, functioning, and heterogeneity (*Baryawno et al., 2019*; *Matsushita et al., 2020*; *Tikhonova et al., 2019*; *Zhong et al., 2020*). The same approaches applied to the periosteum could elicit similar fundamental insights into the regulation and roles of progenitor cells in bone repair mechanisms. These discoveries would also hold high clinical significance and importance as such knowledge could be leveraged to mitigate healing deficiencies seen in about 5–10% of patients, leading to problematic and costly non-unions that are not fully addressed by current therapeutic strategies (*Hak et al., 2014*; *Tzioupis and Giannoudis, 2007*). These discoveries could also be combined from insights stemming from related fields of biomedical research to conceive novel and more potent therapeutic strategies.

One such field of related biomedical research is heterotopic ossification (HO). This pathological process consists of formation and local accumulation of extraskeletal bone (*Meyers et al., 2019*; *Pape et al., 2004*). Its onset and progression are very similar to those occurring during bone fracture repair and involve local inflammation, recruitment of progenitor cells, chondrogenesis and osteogenesis, and deposition of bone. Recent studies from our laboratories and others have revealed that the transforming growth factor-β (TGF-β) superfamily member Activin A has a previously unsuspected role in promoting HO in mice (*Hatsell et al., 2015*; *Hino et al., 2015*; *Mundy et al., 2021*). This protein is best known for its roles in inflammation where it is produced by activated macrophages and regulates cytokine production and profile (*de Kretser et al., 2012*; *Morianos et al., 2019*). The protein has also been found to have pro-chondrogenic activity on human and mouse progenitor cells (*Mundy et al., 2021*; *Djouad et al., 2010*). Given these and other studies, we tested here whether Activin A is a regulator of bone fracture repair as well and may operate as a stimulator of that physiologic process. We demonstrate that local Activin A amounts are markedly increased during fracture healing in mice and patients. A comprehensive scRNA-seq analysis of periosteal mesenchymal lineage cells reveals that the Activin A-encoding gene *Inhba* was expressed in a novel progenitor cell population greatly and rapidly expanding during fracture healing. Systemic immunological interference with endogenous Activin A delayed fracture repair, whereas exogenously provided and locally applied recombinant protein promoted it. These data provide strong support for the important notion that Activin A regulates fracture repair and could offer an effective and easy-to-implement therapeutic tool to enhance it.

## Results

### Amounts and distribution of Activin A increase during fracture repair

To evaluate the participation of Activin A in fracture repair, we first asked whether its presence and distribution were altered during healing, using a standard endochondral tibia fracture model with 2-month-old WT mice. Immunofluorescent staining of intact control tibiae (*Figure 1A*, Intact) showed that Activin A was detectable in only a few cells within the cambium layer of periosteum (*Figure 1Aa*) and in some bone marrow cells (*Figure 1Ab*). Marrow cells normally expressing the Activin A-encoding gene *Inhba* included mesenchymal progenitors and inflammatory cells such as granulocytes,

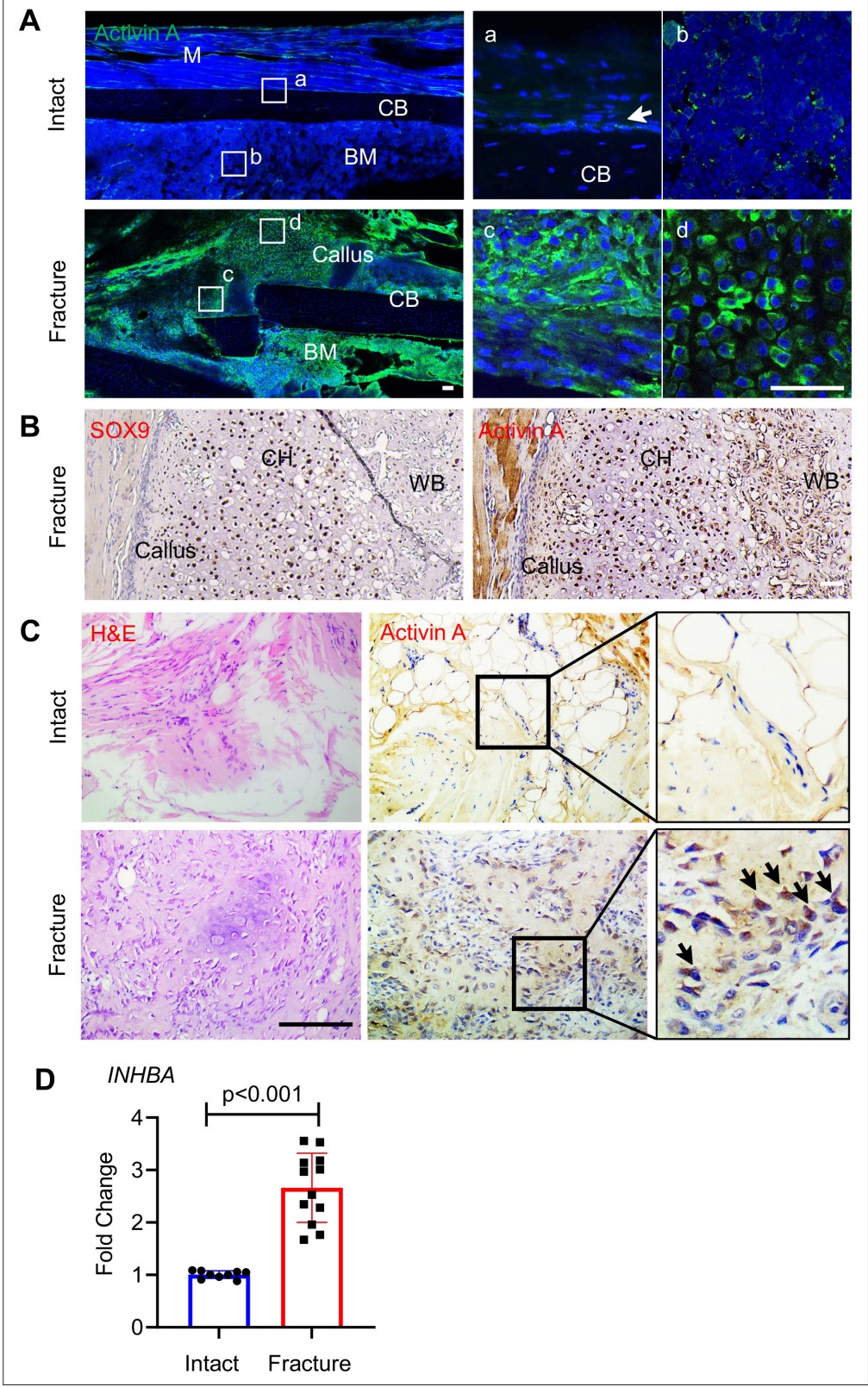

**Figure 1.** Activin A becomes more abundant during fracture repair. (**A**) Whole mount immunofluorescence images of endogenous Activin A distribution in intact and day 5 fractured mouse tibia. The boxed areas in the left panel are shown as enlarged images at the right. M: muscle, CB: cortical bone; BM: bone marrow. Scale bar, 50 μm. (**B**) Immunohistochemistry (IHC) of fracture sections indicates that many Sox9+ chondrocytes within fracture callus

*Figure 1 continued on next page*

*Figure 1 continued*

produce Activin A. CH: chondrocytes; WB: woven bone. Scale bar, 50 µm. (**C**) IHC of human specimens (right panels) shows that Activin A is scanty in intact bone tissue but becomes much more prominent at the fracture site. Left panel: H&E staining. Scale bar, 200 µm. (**D**) Quantative RT-PCR analysis of *INHBA* mRNA in intact and fractured human periosteal tissue samples. *n* = 9–12 specimens/group. Data are expressed as means ± standard deviation (SD) and analyzed by unpaired two-tailed t-test.

The online version of this article includes the following source data and figure supplement(s) for figure 1:

**Source data 1.** Source data for *Figure 1D*.

**Figure supplement 1.** *Inhba* gene expression in mouse bone marrow cell populations.

based on our previous scRNA-seq analysis (*Figure 1—figure supplement 1*; *Zhong et al., 2020*). Five days after fracture, the number and distribution of Activin A-positive cells had dramatically increased (*Figure 1A*, Fracture), and positive cells now included numerous fibroblastic-shaped cells within periosteum (*Figure 1Ac*) and round-shaped early chondrocytes within the developing soft callus (*Figure 1Ad*) co-staining with Sox9 (*Figure 1B*). Immunostaining of surgically retrieval bone specimens from patients showed that Activin A was scanty in connective tissues of intact iliac periosteum (*Figure 1C*, Intact) but abundant at fracture site of long bones (*Figure 1C*, Fracture). Real-Time Quantitative Reverse Transcription PCR Further analysis of these patient specimens showed a nearly threefold increase in *INHBA* mRNA at the fracture site (*Figure 1D*). Clearly, Activin A-producing cells become abundant during fracture repair.

## Fracture callus development involves dynamic cell population shifts

Before testing the possible roles of Activin A in fracture healing directly, we carried out a comprehensive and unbiased analysis of cell populations present during fracture repair using scRNA-seq. Here, we selected Col2a1-Cre;Gt(Rosa)26 tdTomato (Col2/Td) reporter mice because as our previous study indicated, Td+ cells in these mice are major contributors to fracture callus (*Wang et al., 2019*). To establish the effectiveness of this transgenic approach to capture the overall cell populations involved in fracture repair, tibiae from 2-month-old Col2/Td mice were harvested at 5, 10, 15, and 30 days after fracture and processed for spatiotemporal delineation of Td+ cells (*Figure 2—figure supplement 1*). In intact tibiae, the cortical bone surface was covered by a thin layer of periosteum mainly consisting of Td+ cells (*Figure 2—figure supplement 1Aa, Ba*). At day 5 post fracture, Td+ cells had greatly expanded in number to form the thickened periosteum (*Figure 2—figure supplement 1Ab, Bb*). By day 10 when cartilage reached its peak soft callus development as shown by Safranin O staining (*Figure 2—figure supplement 1Bc*), all chondrocytes as well as neighboring fibrotic cells were Td+ (*Figure 2—figure supplement 1Ac*). By day 15 when most cartilage was undergoing endochondral ossification, Td+ cells now constituted the majority of osteoblasts and osteocytes in the callus (*Figure 2—figure supplement 1Ad, Bd*). In the remodeling bone present by day 30, the new periosteum at the edge of callus mostly consisted of Td+ cells (*Figure 2—figure supplement 1Ae, Be*). Colony-forming unit fibroblast (CFU-F) assays of periosteal cells isolated from intact bones revealed that Td+ cells, but not Td– cells, were able to form cell colonies (*Figure 2—figure supplement 1C, D*). Together, the data affirm the fact that the Col2/Td approach captures the overall mesenchymal cell populations taking part in fracture healing.

Having established the effectiveness of the approach, we proceeded to isolate periosteal cells from intact tibiae (termed day 0 cells) and from injured tibiae at days 5 and 10 post fracture and then sorted them for Td+ cells. The percentage of Td+ cells among freshly isolated cells increased from 2.8% at day 0 to 3.4% at day 5 and 8.5% at day 10 (*Figure 2—figure supplement 2A*). Using the 10× Genomics approach, we sequenced 7,496, 7,535, and 10,398 Td+ cells from days 0, 5, and 10 samples, respectively, with a median of 3,462 genes/cell and 13,463 unique molecular identifiers (UMIs)/cells (*Figure 2—figure supplement 2B*). We merged the resulting three datasets comprising an overall total of 25,429 cells that resolved into 16 cell clusters, including 6 clusters of periosteal mesenchymal lineage cells, 4 clusters of hematopoietic cells, 1 cluster of muscle cells, 1 cluster of synovial lining cells, 1 cluster of tendon cells, 1 cluster of endothelial cells (ECs), 1 cluster of Schwann cells, and 1 cluster of smooth muscle cells (*Figure 2—figure supplement 2C, D*; *Supplementary file 1a*). Td expression was detected in all clusters but was highest in mesenchymal cells (*Figure 2—figure supplement 2E*). There was basal Td expression in non-mesenchymal cells, which was also observed

in previous scRNA-seq studies of bone marrow mesenchymal lineage cells from our group and others (*Baryawno et al., 2019*; *Matsushita et al., 2020*; *Tikhonova et al., 2019*; *Zhong et al., 2023*). Heatmap analysis revealed the hierarchy and diverse nature of these cell clusters with distinct gene signatures (*Figure 2—figure supplement 2F*).

The presence of synovial fibroblasts and tenocytes in our datasets, exclusively existing in days 0 and 5 samples, might be due to insufficient agarose coverage of the two ends of tibia during the cell enzymatic digestion step of intact and fractured bones. Day 10 samples did not have these two clusters because only callus was dissected out for scRNA-seq analysis (*Supplementary file 1a*). Given the focus of our study on mesenchymal lineage cells in callus, we digitally removed those clusters as well as other non-mesenchymal cells. As a result, the recalculated data identified six distinct mesenchymal cell clusters (*Figure 2A, B*; *Supplementary file 1b*). Based on lineage-specific traits, clusters 3, 4, 5, and 6 represented early osteoblasts (EOB), osteoblasts (OB), chondrocytes (CH), and hypertrophic chondrocytes (HCH), respectively (*Figure 2C*). Cells in cluster 1 were characterized by several typical stem cell markers such as *Cd34*, *Ly6a* (Sca1), and *Thy1* (Cd90), suggesting that they represented mesenchymal progenitor cells (MPCs). Cells constituting the expansive cluster 2 expressed the above stem cell markers at a very low level and expressed lineage-specific gene markers such as those of OBs and CHs at a low level as well. When the merged datasets (*Figure 2A*) were separated by time point, it became clear that cluster 2 cells markedly increased in number early from days 0 to 5, whereas chondrocytes (cluster 5), hypertrophic chondrocytes (cluster 6), and osteoblasts (clusters 3 and 4) expanded by day 10 (*Figure 2B* and *Figure 2—figure supplement 3*). Interestingly, computational cell cycle analysis revealed that cluster 2 contained highly proliferative cells, particularly at day 5 (*Figure 2D, E*). Several proliferation marker genes, such as *Ccnd3*, *Cdk4*, *Cdc20*, *Cdca3*, *Mcm4*, and *Cepna*, were highly expressed in cluster 2 at the day 5 time point (*Figure 2—figure supplement 4*). Thus, we termed – and refer to – cluster 2 cells as proliferative progenitor cells (PPCs).

Examination of previously reported markers of periosteal mesenchymal progenitors in our dataset revealed that they were either ubiquitously or specifically expressed amongst the cell populations (*Figure 2—figure supplement 5A*). *Ly6a* and *Thy1* (*Matthews et al., 2021*) were largely restricted to MPCs, and *Acta2* (*Grcevic et al., 2012*) demarcated the PPC population. Genes such as *Itgb1* (Cd29) (*Duchamp de Lageneste et al., 2018*) were expressed at high level in all clusters, whereas *Gli1* and *Lepr* (*Shi et al., 2017*; *Shu et al., 2021*) were expressed at very low levels. *Itgav* (Cd51), *Eng* (Cd105), *Ctsk*, *Postn*, *Prxx1*, *Pdgfra*, and *Pdgfrb* (*Duchamp de Lageneste et al., 2018*; *Böhm et al., 2019*; *Chan et al., 2015*; *Debnath et al., 2018*; *Esposito et al., 2020*; *Julien et al., 2022*; *Marecic et al., 2015*) were prominent in progenitors (MPCs and PPCs) but low in mature cells. Cd200 (*Chan et al., 2015*; *Marecic et al., 2015*) was expressed at low levels in MPCs but its expression was higher in more mature cells. *Col2a1* (*Wang et al., 2019*) and *Sox9* (*He et al., 2017*) were prominent in chondrocytes compared to other populations, consistent with their being well-established cartilage markers. We did not detect the expression of a previously proposed mesenchymal cell marker *Mx1* (*Ortinau et al., 2019*) in our datasets.

RNA velocity delineates cellular differentiation paths and transient phenotypic states from scRNA-seq data (*Bergen et al., 2020*). Applying this approach to our merged datasets above (*Figure 2A*), we found that directionality of cell differentiation and diversification started from cluster 1 (MPCs), advanced and transitioned through cluster 2 (PPCs) and ended in cluster 4 (OBs) and cluster 6 (HCH) (*Figure 2—figure supplement 6*). Likewise, at every time point, pseudotemporal cell trajectory analysis placed MPC cells (cluster 1) at one end of the developmental trajectory, PPC cells (cluster 2) in a central position, and OBs (cluster 4) and CHs (clusters 5 and 6) at the other two ends (*Figure 2F*).

Together, the data strongly indicate that MPCs serve as stem/progenitors and give rise to PPCs which in turn diverge into chondrocytes and osteoblasts, contributing to soft and hard callus formation.

## PPCs strongly express *Inhba* and gain myofibroblast-like features after fracture

Given the apparent developmental centrality of the PPC population, we sought to characterize it further by defining their differentially expressed genes (DEGs) compared to those in the other cell clusters, using GO term analyses (*Figure 3A*). In the merged dataset, the most up-regulated genes in PPCs indicated their myofibroblast-like phenotype. Those genes were closely related to processes and pathways known to be regulated by myofibroblasts such as wound healing, focal adhesion,

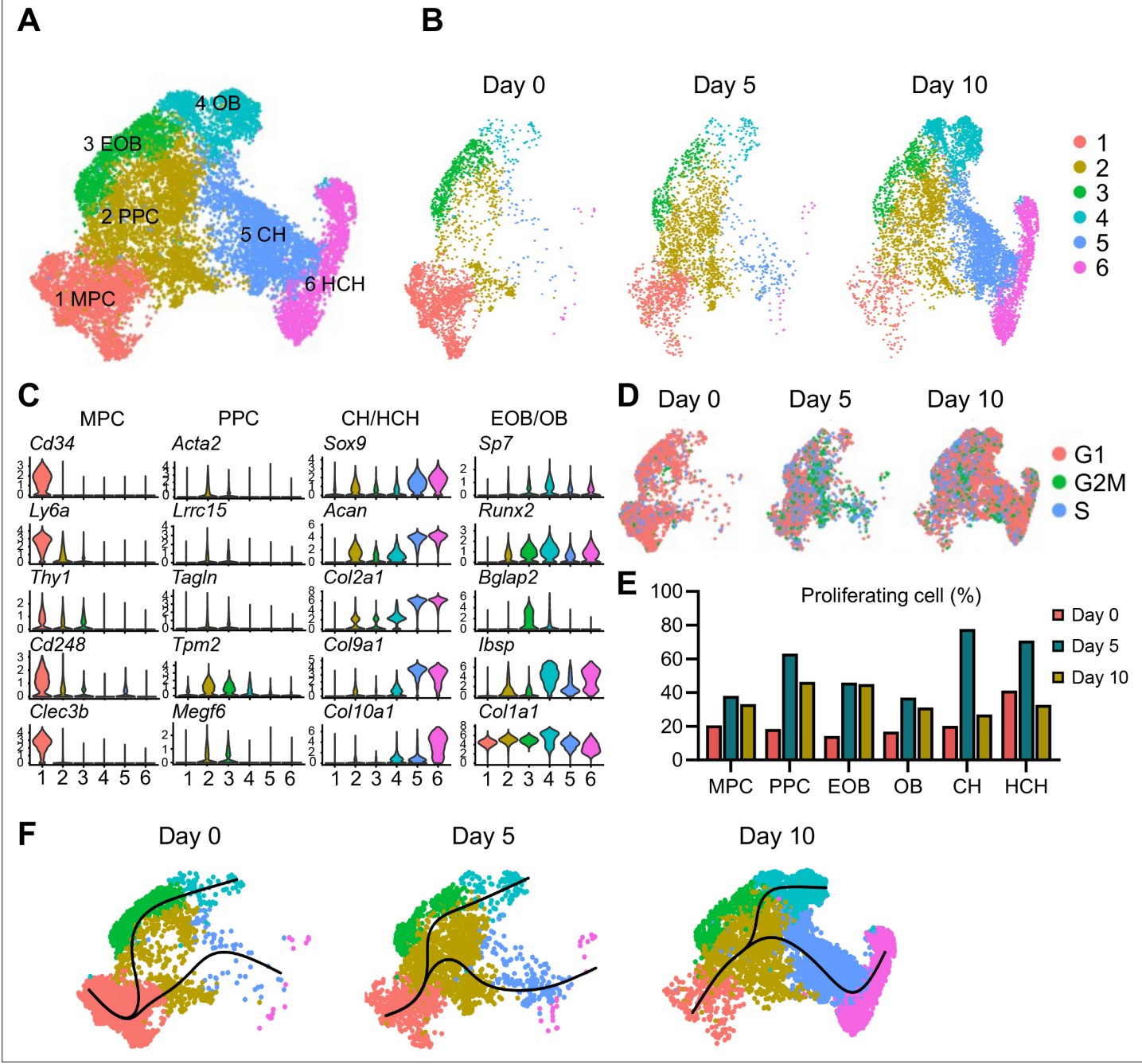

**Figure 2.** Single-cell transcriptomics analyses reveal identities and developmental trajectories of periosteal mesenchymal populations. (**A**) The uniform manifold approximation and projection (UMAP) plot of 13,040 Td+ mesenchymal lineage cells isolated from tibial periosteum of 2-month-old Col2/Td mice. Datasets from cells isolated from intact periosteum (day 0) and fracture site on days 5 and 10 post-surgery were merged and combined into a single plot. (**B**) UMAP plots of those cells shown at individual time point. (**C**) Violin plots of cluster-specific makers of mesenchymal lineage cells. MPC: mesenchymal progenitor cell; PPC: proliferative progenitor cell; CH: chondrocyte; HCH: hypertrophic chondrocyte; EOB: early osteoblast; OB: osteoblast. (**D**) Cell cycle phase of periosteal mesenchymal lineage cells at days 0, 5, and 10. (**E**) The percentage of proliferative cells (S/G2/M phase) in each cell cluster at days 0, 5, and 10 was computationally quantified. (**F**) Slingshot trajectory plots of periosteum mesenchymal lineage cells at days 0, 5, and 10.

The online version of this article includes the following source data and figure supplement(s) for figure 2:

**Source data 1.** Source data for *Figure 2E*.

**Figure supplement 1.** Col2/Td labels periosteal mesenchymal progenitors in intact and fractured tibiae.

**Figure supplement 1—source data 1.** Source data for *Figure 2—figure supplement 1C*.

*Figure 2 continued on next page*

*Figure 2 continued*

**Figure supplement 2.** Large-scale single-cell RNA-sequencing (scRNA-seq) analysis of Td+ cells sorted from tibial periosteum of 2-month-old Col2/Td mice.

**Figure supplement 3.** PPCs are greatly and rapidly expanded after fracture.

**Figure supplement 3—source data 1.** Source data for *Figure 2—figure supplement 3A*.

**Figure supplement 3—source data 2.** Source data for *Figure 2—figure supplement 3B*.

**Figure supplement 4.** Violin plots of proliferation makers in mesenchymal lineage cells within the single-cell RNA-sequencing (scRNA-seq) datasets.

**Figure supplement 5.** The expression patterns of previously reported periosteal mesenchymal progenitor markers are shown in uniform manifold approximation and projection (UMAP) plots.

**Figure supplement 6.** RNA velocity analysis predicts differentiation routes of periosteal mesenchymal cells during fracture healing.

extracellular matrix organization, and contractile actin filament (*Hsia et al., 2016*), and included known myofibroblast marker genes such as *Acta2*, *Tagln*, *Tagln2*, *Myl9*, *Actg1*, *Tpm2*, and *Fbn2* (*Figure 3B* and *Figure 3—figure supplement 1A*; *Hsia et al., 2016*; *López-Antona et al., 2022*). The expression of these genes was highly up-regulated at day 5 and then reduced at day 10 (*Figure 3B*), suggesting that fracture transiently promotes PPCs into a myofibroblast-like phenotype. Conversely, the least expressed genes were those related to bone mineralization and chondrocyte differentiation, confirming that the PPCs did not possess a terminally differentiated phenotype (*Figure 3A, B*). Particularly relevant to the present study was the finding that compared to MPCs, PPCs highly expressed *Inhba* after fracture (*Figure 3C* and *Figure 3—figure supplement 1B*). Note that chondrocytes also highly expressed *Inhba*, consistent with the immunostaining results shown in *Figure 1*. However, their number was much lower than PPCs in early callus (*Figure 2—figure supplement 3B*), suggesting that PPCs are likely to be the main source of Activin A in early fracture healing. Activin A binds to type II receptors (ActRIIA or ActRIIB) to recruit and phosphorylate type I receptors (ALK4 or ALK7) for initiating its intracellular signaling (*Pangas and Woodruff, 2000*). uniform manifold approximation and projection (UMAP) plots suggested that genes encoding these receptors (*Acvr2a*, *Acvr2b*, *Acvr1b*, and *Acvr1c*, respectively) were expressed in all mesenchymal progenitor populations and *Acvr2a* expression was enriched in MPCs (*Figure 3—figure supplement 1C*).

To validate the above findings, we focused on the interval between days 0 and 5 when the PPCs increased the most in number (*Figure 2B*). Based on scRNA-seq data (*Figure 2—figure supplement 5B*), we sorted Cd45−Cd31−Ter119−Cd34+ (Lin−/Cd34+) cells and Cd45−Cd31−Ter119−Cd34− (Lin−/Cd34−) cells to represent MPCs and PPCs, respectively. Note that Lin−Cd34− cells also include more mature cells, such as osteoblasts and chondrocytes, but their number is much lower than PPCs at day 5 (*Figure 2—figure supplement 3B*). EdU incorporation indicated that MPCs were less proliferative than PPCs at both days 0 and 5, though bone fracture did enhance proliferation in both populations (*Figure 3D*). In addition, quantitative RT-PCR (qRT-PCR) analysis of cells sorted from day 5 callus verified that Cd34+ cells highly expressed MPC markers including *Cd34*, *Ly6a*, *Cd248*, and *Clec3b*, whereas Cd34− cells more strongly expressed myofibroblast markers (*Acta2* and *Tagln*) as well as *Inhba* (*Figure 3E*). Lastly, immunohistochemistry (IHC) on day 5 fractures from Col2/Td mice revealed that many Td+ cells were positive for both αSMA and Activin A (*Figure 3F*). Those double positive cells included not only progenitors (*Figure 3Fa*) but also early chondrocytes (*Figure 3Fb*). Together, the data above provide further evidence for the occurrence of MPCs and PPCs within the evolving fracture callus and validate the myofibroblast-like phenotype of PPCs characterized also by *Inhba* expression after fracture.

## Activin A stimulates proliferation and differentiation in periosteal progenitors

The spatiotemporal links between PPC expansion and *Inhba* expression during fracture repair progression above indicated that Activin A may directly promote progenitor cell proliferation and differentiation. To test this possibility, we cultured tibial periosteal mesenchymal progenitors and treated them with recombinant Activin A (100 ng/ml) or with a neutralizing monoclonal antibody against mouse Activin A (nActA.AB; 100 μg/ml) that we used in a previous study (*Mundy et al., 2021*). Cell number analysis on day 3 indicated that Activin A treatment did stimulate cell proliferation, whereas treatment with nActA.AB inhibited it (*Figure 4A*). Remarkably, Activin A treatment up-regulated the

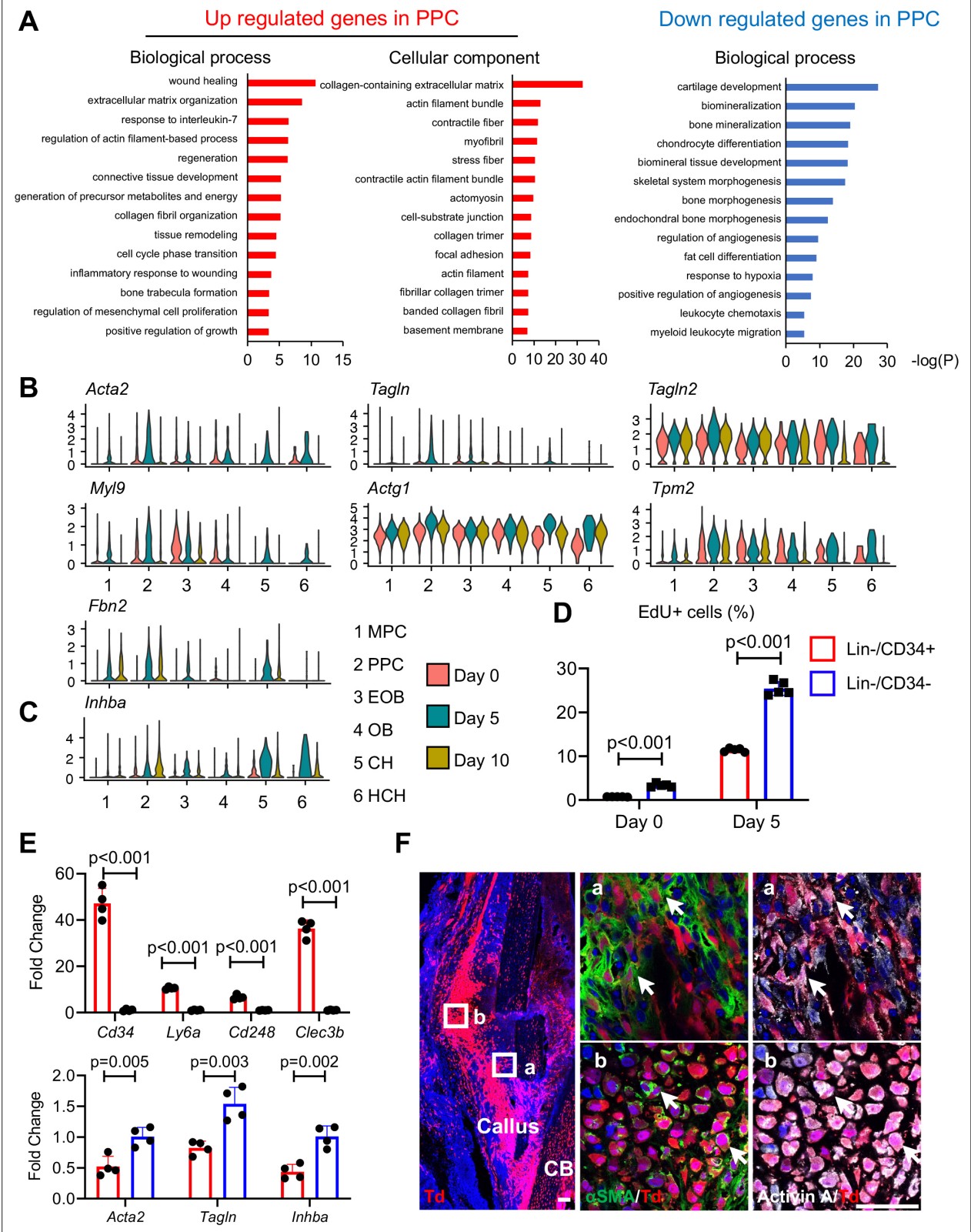

**Figure 3.** Proliferative progenitor cells (PPCs) have a myofibroblast-like phenotype and express *Inhba*. (**A**) GO term analysis of genes up- or down-regulated in the PPCs (cluster 2) compared to other periosteal mesenchymal cell clusters. (**B**) Violin plots of myofibroblastic cell marker gene expression. (**C**) Violin plots of *Inhba* gene expression. (**D**) Flow analysis of EdU+ cells in mesenchymal progenitor cells (MPCs; Lin−Cd34+) and PPCs (Lin−Cd34−) from the periosteum of intact and fractured (day 5) mouse bones. *n* = 4 mice/group. (**E**) qRT-PCR analyses of stem cell markers (top), myofibroblast

*Figure 3 continued on next page*

Figure 3 continued

markers, and *Inhba* (bottom) in MPCs and PPCs at day 5 post fracture. *n* = 4 mice/group. Data are expressed as means ± SD and analyzed by unpaired two-tailed t-test. (**F**) Whole mount immunofluorescence images of αSMA and Activin A distribution in mouse callus at day 5 post fracture. Boxed areas in the left panel are shown enlarged on the right. Arrows point to representative Td+ cells that are co-stained with both αSMA and Activin A antibodies. CB: cortical bone. Scale bar, 50 μm.

The online version of this article includes the following source data and figure supplement(s) for figure 3:

**Source data 1.** Source data for *Figure 3D*.

**Source data 2.** Source data for *Figure 3E*.

**Figure supplement 1.** The expression patterns of myofibroblast-like cell markers, *Inhba*, and its receptors in fracture healing.

levels of expression of myofibroblastic cell markers including αSMA/*Acta2* (*Figure 4B, C*) and *Tagln* (*Figure 4C*) as did treatment with recombinant TGF-β1 which is known for its ability to promote myofibroblast development (*Wynn, 2008*). Next, we tested whether Activin A was able to directly stimulate chondrogenic and osteogenic cell differentiation that as predicted by trajectory analysis (*Figure 2F*). Periosteal cell cultures reared in basal chondrogenic or osteogenic media were treated with Activin A as above. The treatment did stimulate chondrogenesis versus control cultures as revealed by strong alcian blue staining and higher expression of such cartilage markers as *Col2a1*, *Acan*, and *Sox9* (*Figure 4D, E*). Activin A treatment did not appreciably enhance osteogenic differentiation (*Figure 4F, G*). However, nActA.AB treatment did inhibit both osteogenesis and chondrogenesis (*Figure 4D–G*). Thus, endogenous and exogenous Activin A acts to promote chondrogenic and osteogenic differentiation in periosteal progenitors.

## Systemic administration of Activin A neutralizing antibody delays fracture repair

Given the apparent ability of Activin A to stimulate periosteal progenitor cell proliferation and differentiation, it became reasonable to predict that the protein would have a positive and important role in fracture repair. To test this thesis, we subjected 2-month-old WT mice to the same closed tibia fracture injury as above that heals via endochondral ossification. The animals were randomly divided into two groups. The first group received biweekly subcutaneous injections of nActA.AB [immunoglobulin G2b (IgG2b) isotype at 10 mg/kg per injection] as in our previous study. The second group served as control and received injections of pre-immune IgG2b isotype antibody obtained from the same manufacturer and given at identical dose, route, and frequency. Based on the spatiotemporal patterns of fracture healing in this model (*Figure 2—figure supplement 1*), tibias from each group were harvested at 5, 7, 10, and 14 days after surgery to capture and analyze the cartilage and bone formation phases and at 6 weeks to measure the ultimate effectiveness of bone healing by mechanical tests. Histochemical analysis clearly showed that nActA.AB administration significantly reduced overall callus size and cartilage and bone areas at all time points post fracture (5, 7, 10, and 14 days) compared to isotype antibody controls (*Figure 5A, B*). The changes in callus volume and bone volume were verified by micro-computed tomography (μCT) analysis (*Figure 5—figure supplement 1*). By 6 weeks in the control group, the tibial fractures were all bridged, indicating a successful recovery but those in the nActA.AB treatment group were lagging, leading to a decrease in fracture healing score (*Figure 5C*, p = 0.006). Furthermore, three-point bending analysis revealed 44.4%, 35.0%, and 29.0% reductions in energy to failure, stiffness, and peak load, respectively, in fractured tibias from nActA. AB- versus isotype-treated mice (*Figure 5D*), all statistically significant.

To gain insights into whether systemic nActA.AB administration affected PPCs, we performed qRT-PCR and immunostaining analyses on early fracture samples from wild-type mice as above. At day 7 post fracture, nActA.AB administration reduced the number of cells positive for phosphorylated SMAD2 (pSMAD2) through which Activin A normally signals intracellularly (*Pangas and Woodruff, 2000*), suggesting the effectiveness of neutralizing antibody treatment (*Figure 5E, F*). Interestingly, the number of PPCs positive for αSMA and the percentage of pSMAD2+ cells within PPC population were significantly decreased, while the percentage of pSMAD2+ cells within non-PPCs remained the same. These data were further con firmed by reduced gene expression of *Acta2* and *Inhba* in fracture callus after nActA.AB administration (*Figure 5G*). Taken together, our

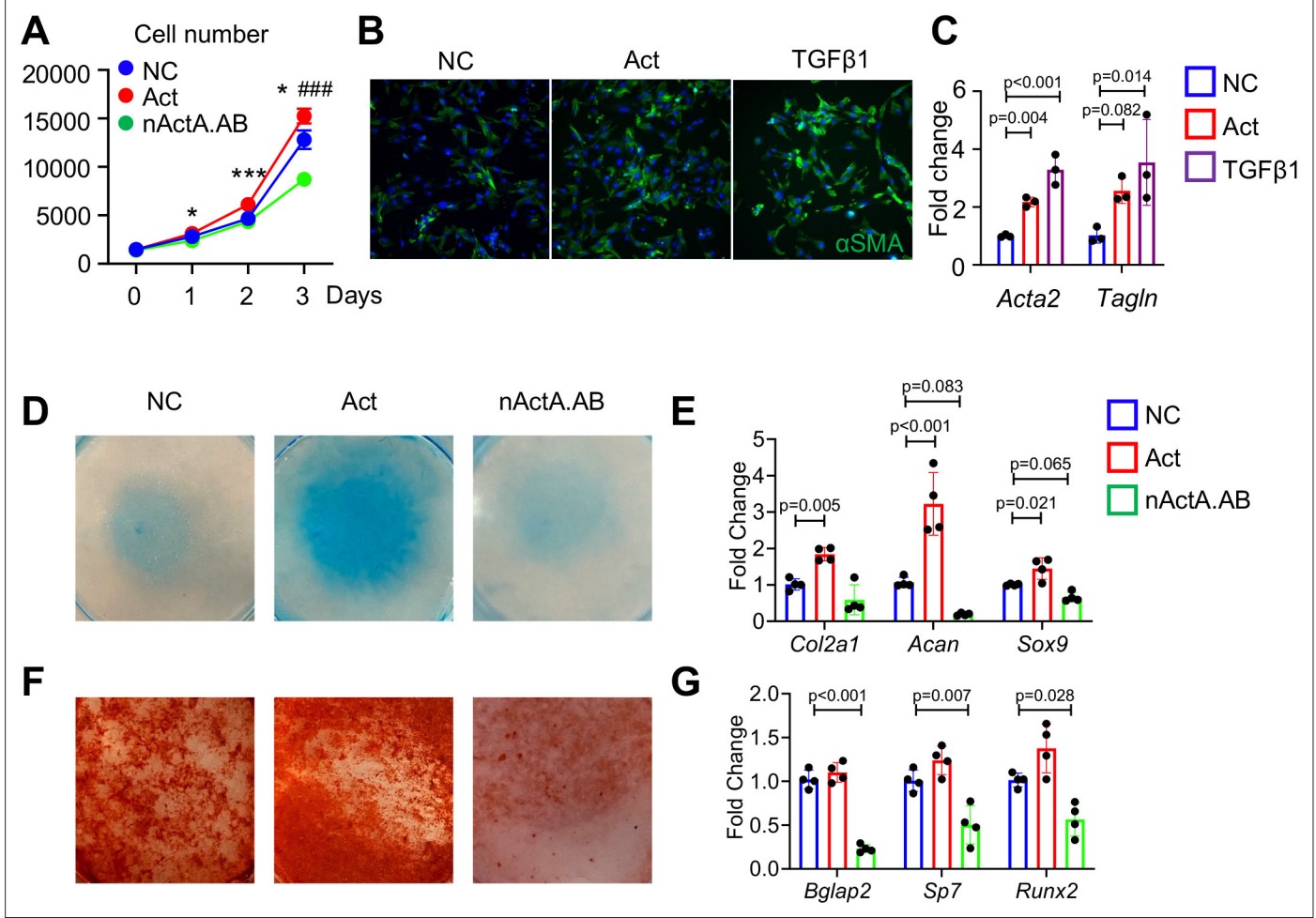

**Figure 4.** Activin A regulates proliferation and differentiation of periosteal mesenchymal progenitors in vitro. (**A**) Proliferation assay of periosteal mesenchymal progenitors treated with recombinant Activin A (Act) or neutralizing monoclonal antibody (nActA.AB) versus control antibody (NC). * p<0.05, *** p<0.001 Act vs NC. ### p<0.001 nActA.AB vs NC. (**B**) Immunofluorescence images of αSMA in periosteal mesenchymal progenitors treated with Activin A (100 ng/ml) or TGF-β1 (10 ng/ml) for 3 days. (**C**) qRT-PCR analysis of myofibroblast-like marker expression. (**D**) Alcian blue staining of periosteal mesenchymal progenitors in micromass cultures undergoing chondrogenic differentiation in the presence of Activin A (100 ng/ml) or nActA. AB (100 μg/ml) for 2 weeks. (**E**) qRT-PCR analyses of chondrogenic markers. (**F**) Alizarin red staining of periosteal mesenchymal progenitors undergoing osteogenic differentiation in the presence of Activin A or nActA.AB for 2 weeks. (**G**) qRT-PCR analyses of osteogenic markers. Data are expressed as means ± SD and analyzed by one-way ANOVA with Tukey post-hoc test.

The online version of this article includes the following source data for figure 4:

**Source data 1.** Source data for *Figure 4A*.

**Source data 2.** Source data for *Figure 4C*.

**Source data 3.** Source data for *Figure 4E*.

**Source data 4.** Source data for *Figure 4G*.

results clearly suggest that the PPCs were the primary responsive cell type to Activin A in early fracture and that systemic interference of Activin A action by nActA.AB treatment impaired fracture healing.

We also examined contralateral, uninjured tibiae in all mice above and asked whether Activin A normally regulates bone homeostasis. μCT scanning showed that trabecular and cortical bone structure was essentially identical in nActA.AB- and control isotype-treated mice (*Figure 5—figure supplement 2*), indicating that Activin A does not have major homeostatic roles, at least within the time frame of our studies.

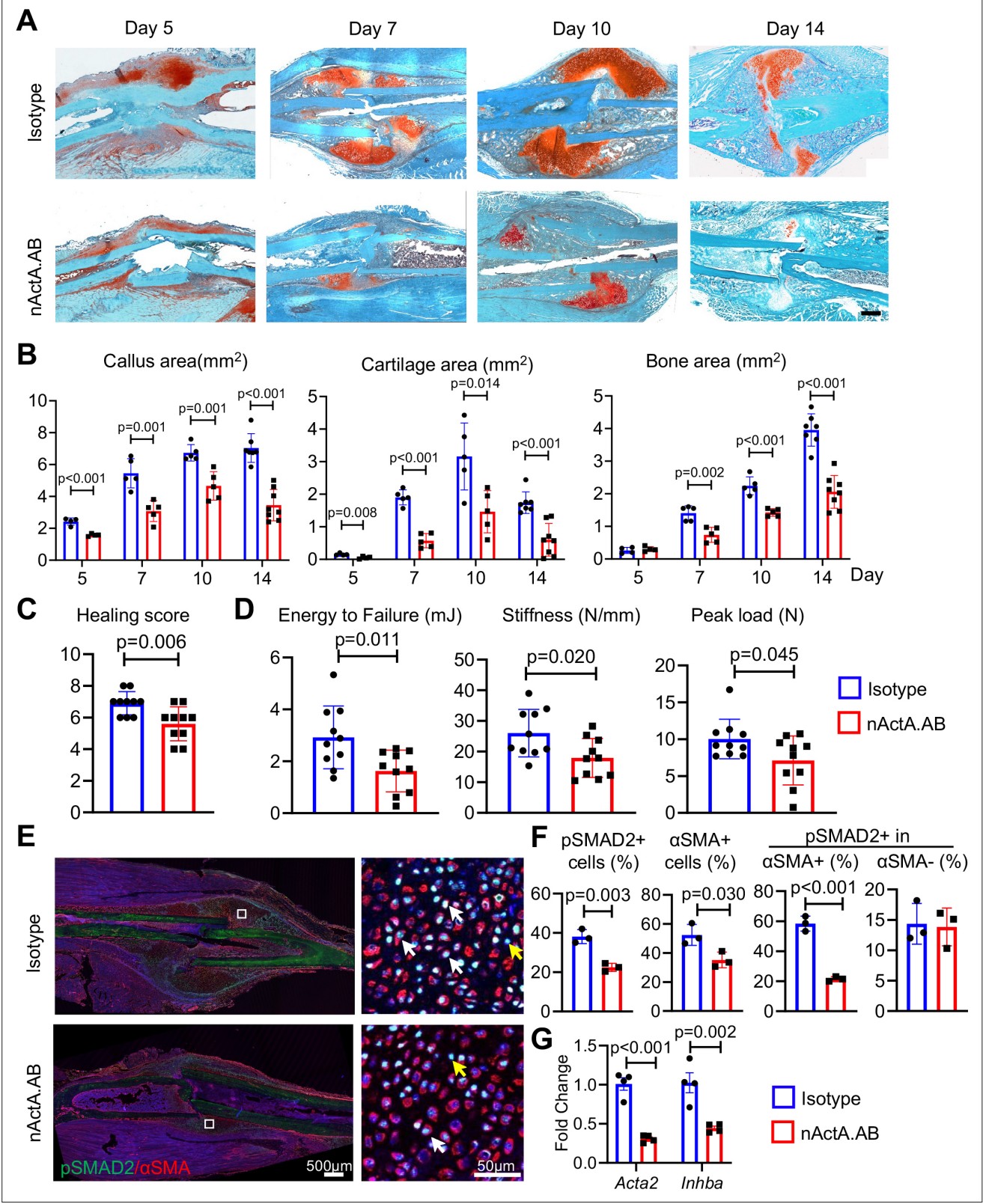

**Figure 5.** Systemic administration of Activin A antibody delays mouse fracture healing. (**A**) Representative Safranin O/Fast green staining images of fracture calluses at days 5, 7, 10, and 14 post fracture. Mice received subcutaneous injections of control IgG2b isotype or neutralizing monoclonal antibody against Activin A (nActA.AB, 10 mg/kg) twice a week after fracture. Scale bar, 1 mm. (**B**) Callus area, cartilage area, and bone area were quantified at indicated time points. *n* = 4–7 mice/time point. (**C**) Measurement of fracture healing scores at 6 weeks post fracture. *n* = 10 mice/group.

*Figure 5 continued on next page*

*Figure 5 continued*

(**D**) Mechanical testing was performed on bones at 6 weeks post fracture. *n* = 10 mice/group. (**E**) Immunofluorescence images of pSMAD2 and αSMA in fracture calluses of control (isotype) and nActA.AB-treated mice at day 7 post fracture. White arrows point to pSMAD2+αSMA+ cells and yellow arrows point to pSMAD2+αSMA− cells. Scale bar, 500 μm (low mag), 50 μm (high mag). (**F**) Percentages of pSMAD2+ and αSMA+ cells in fracture calluses and pSMAD2+ cells within the αSMA+/− populations were quantified. *n* = 3 mice/group. (**G**) qRT-PCR analyses of *Acta2* and *Inhba* expression in day 7 callus from 2-month-old mice treated with nActA.AB versus isotype control. *n* = 4 mice/group. Data are expressed as means ± SD and analyzed by unpaired two-tailed t-test.

The online version of this article includes the following source data and figure supplement(s) for figure 5:

**Source data 1.** Source data for *Figure 5B*.

**Source data 2.** Source data for *Figure 5C*.

**Source data 3.** Source data for *Figure 5D*.

**Source data 4.** Source data for *Figure 5F*.

**Source data 5.** Source data for *Figure 5G*.

**Figure supplement 1.** Immunological inhibition of Activin A impedes fracture healing.

**Figure supplement 1—source data 1.** Source data for *Figure 5—figure supplement 1B*.

**Figure supplement 2.** Immunological inhibition of Activin A does not affect normal bone homeostasis.

**Figure supplement 2—source data 1.** Source data for *Figure 5—figure supplement 2B*.

**Figure supplement 2—source data 2.** Source data for *Figure 5—figure supplement 2D*.

## Local supplementation of recombinant Activin A accelerates fracture healing

To extend the above studies, we carried out complementary studies asking whether exogenous Activin A would enhance fracture healing and could thus represent a potential therapeutic. As above, we used a closed tibial fracture model with 2- and 20-month-old mice since older mice are more clinically relevant when testing a potential therapy. Immediately after fracture, a 50-μl aliquot of Matrigel containing recombinant Activin A was microinjected at the operated site; controls received Matrigel alone. Mice were then harvested at 5, 14, and 28 days from fracture to monitor the healing process. Notably, exogenous Activin A implantation had clearly increased callus size and cartilage and bone areas at each time point and in both age groups, based on histochemistry (*Figure 6A, B*) and μCT imaging (*Figure 6—figure supplement 1*). IHC revealed that Activin A implantation had elicited a major increase in the number of mesenchymal cells positive for pSMAD2 and αSMA and the percentage of pSMAD2+ cells within αSMA+ population (but not within αSMA− population) in fracture calluses of young mice at day 5 post fracture (*Figure 6C, D*). Importantly, overall healing scores were significantly increased in both young and old mice at 6 weeks post fracture (*Figure 6E*). Mechanical testing revealed that energy to failure, stiffness, and peak load were all significantly increased by Activin A implantation in 20-month-old mice (*Figure 6F*). In line with previous reports (*Liu et al., 2022*), young mice had stronger bone than old mice with increased stiffness and peak load. We also noted a trend of increase in energy to failure in young mice as well after Activin A treatment but was not statistically significant. Taken together, the data indicate a quick expansion of mesenchymal progenitors and a promotion of healing following Activin A supplementation.

## Activin A promotes intramembranous bone defect repair

To strengthen our observations, we carried out an additional set of loss- and gain-of-function experiments using a unicortical non-critical size (0.8 mm) drill-hole bone repair model that heals mainly through intramembranous ossification (*Minear et al., 2010*). In this model, woven bone formation is usually observed by day 7 post-surgery, and bone bridging and re-corticalization occur by day 21 (*Liu et al., 2019*). Accordingly, drill-hole surgery was carried out on femoral mid-shaft of 2-month-old mice. For loss-of-function tests, mice were given biweekly subcutaneous injections of nActA.AB or isotype control as above. For gain-of-function tests, a 50-μl aliquot of Matrigel containing up to 1 μg of recombinant Activin A was microinjected inside the medullary canal at the drill site; Matrigel alone was microinjected in companion controls. In all controls, bone formation became evident in the medullary region of interest by day 7 post-surgery and extensive bone formation in the drilled region had occurred by day 21 (*Figure 7A*). Nearly all day 21 samples displayed complete defect bridging

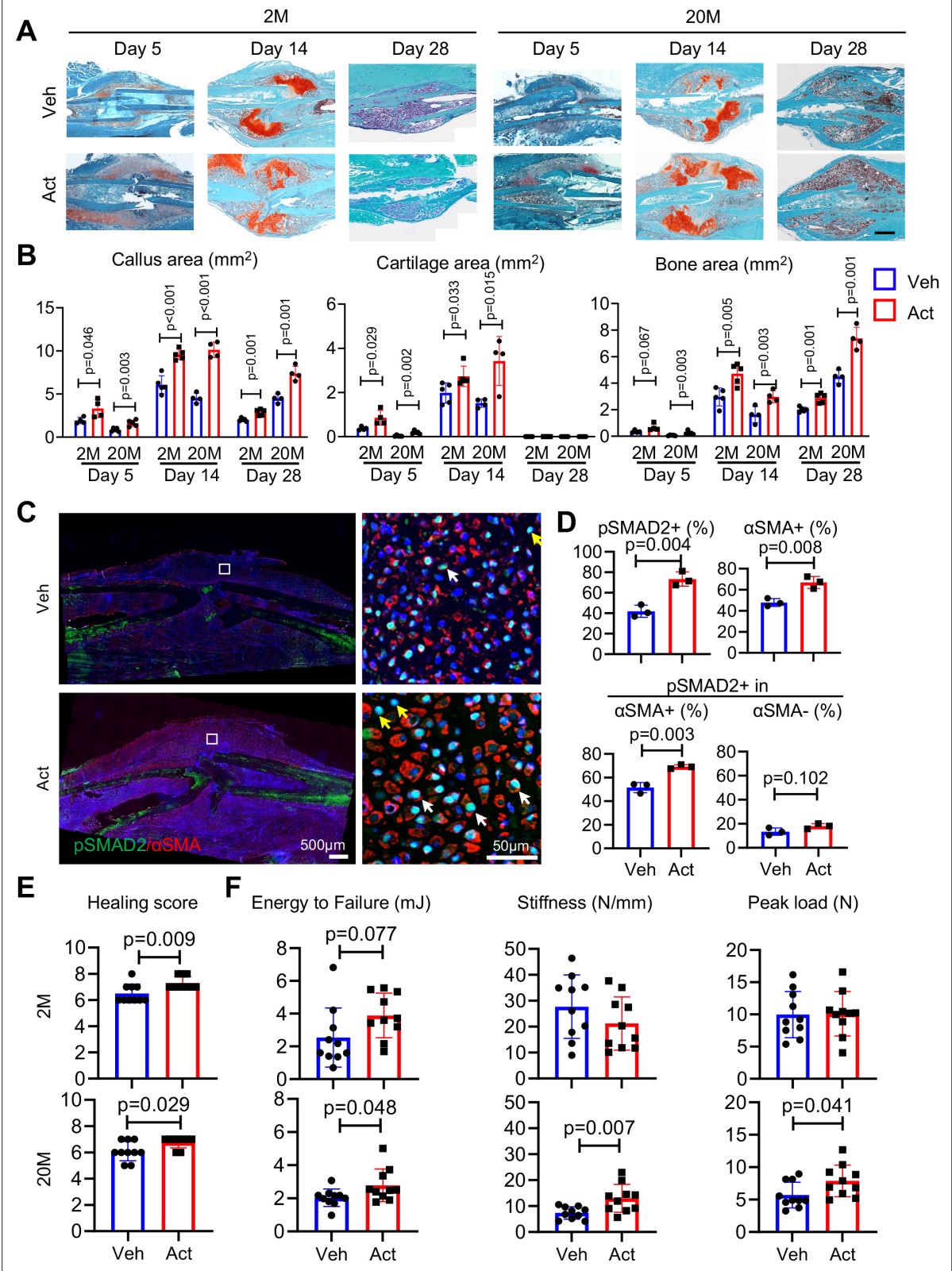

**Figure 6.** Local Activin A implantation promotes fracture healing. (**A**) Representative Safranin O/Fast green staining histochemical images of fracture calluses at days 5, 14, and 28 post fracture. Two-month-old (2M) or 20-month-old (20M) mice were implanted with a 50-µl Matrigel aliquot containing vehicle (Veh) or Activin A (ACT) (1 µg) at the fracture site immediately after surgery. Scale bar, 1 mm. (**B**) Callus area, cartilage area, and bone area were quantified at indicated time points. $n$ = 4 mice/group. (**C**) Immunofluorescence images of pSMAD2 and αSMA in fracture calluses of control and

*Figure 6 continued on next page*

*Figure 6 continued*

ACT-implanted mice at day 5 post fracture. White arrows point to pSMAD2+αSMA+ cells and yellow arrows point to pSMAD2+αSMA– cells. Scale bar, 500 μm (low mag), 50 μm (high mag). (**D**) Percentage of pSMAD2+, αSMA+ cells in fracture calluses and pSMAD2+ cells within αSMA+/– populations were quantified. *n* = 3 mice/group. (**E**) Fracture healing scores were quantified in bones of 2- and 20-month-old mice at 6 weeks post fracture. *n* = 10 mice/group. (**F**) Mechanical testing was performed on bones of 2- and 20-month-old mice at 6 weeks post fracture. *n* = 10 mice/group. Data are expressed as means ± SD and analyzed by unpaired two-tailed t-test.

The online version of this article includes the following source data and figure supplement(s) for figure 6:

**Source data 1.** Source data for *Figure 6B*.

**Source data 2.** Source data for *Figure 6D*.

**Source data 3.** Source data for *Figure 6E*.

**Source data 4.** Source data for *Figure 6F*.

**Figure supplement 1.** Activin A treatment accelerates fracture healing.

**Figure supplement 1—source data 1.** Source data for *Figure 6—figure supplement 1B*.

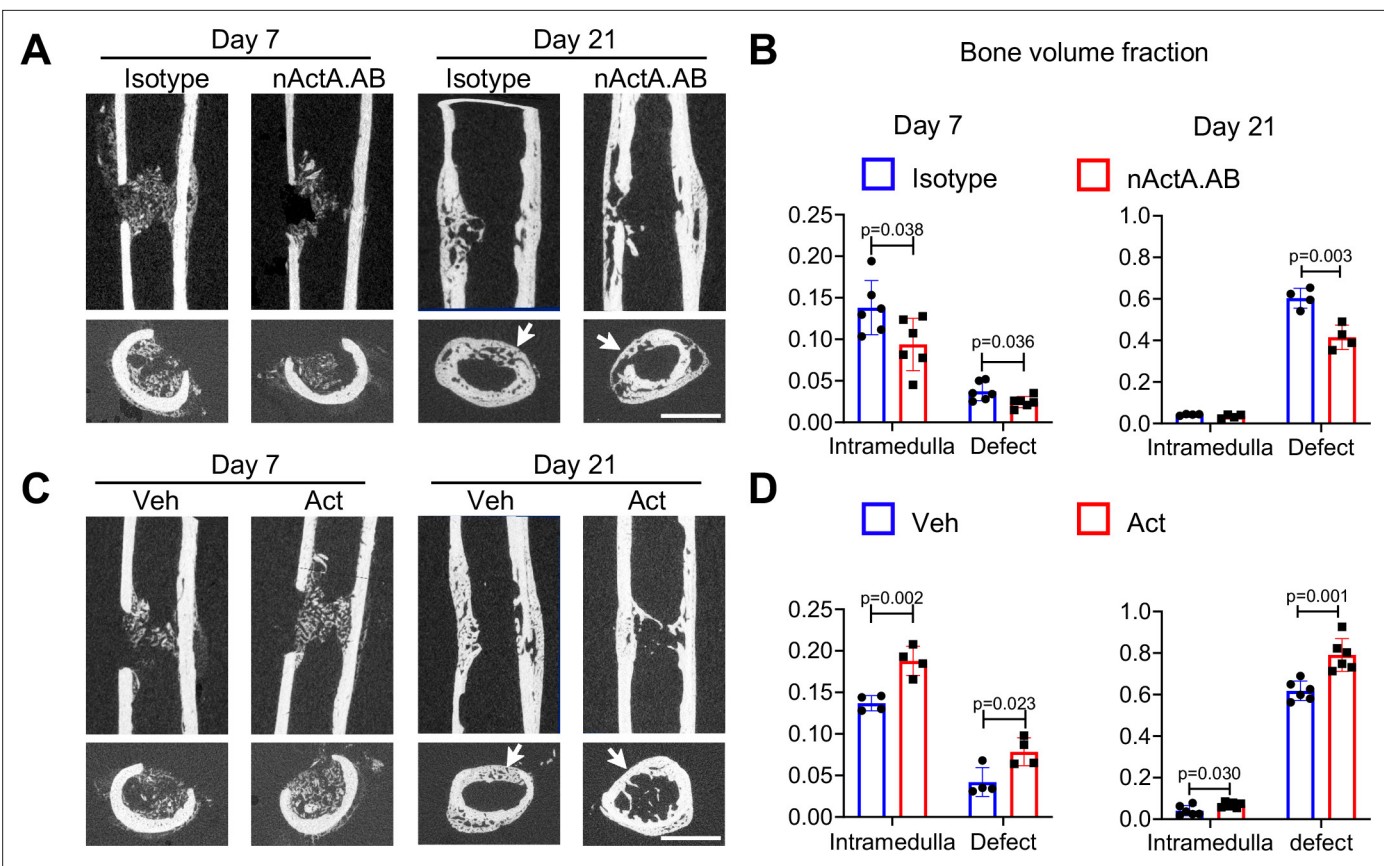

**Figure 7.** Activin A promotes intramembranous bone repair of unicortical drill holes. (**A**) Representative sagittal (top) and transverse (bottom) cross-sections of micro-computed tomography (μCT) images of Activin A blocking antibody (nActA.AB)-treated drill-hole defects. Mice received injections of control IgG2b isotype or neutralizing monoclonal antibody against Activin A (nActA.AB, 10 mg/kg) twice a week after drill-hole injury. Arrows point to the defect region. Scale bar, 1 mm. (**B**) Bone volume fraction of intramedullary and cortical defect regions at days 7 and 21 post-injury. *n* = 4–6 mice/group. (**C**) Representative sagittal (top) and transverse (bottom) cross-sections of μCT images of recombinant Activin A (Act)-treated drill-hole defects. Two-month-old mice were implanted with a 50-μl Matrigel aliquot containing vehicle (Veh) or Activin A (Act) (1 μg) at the drill-hole site immediately after surgery. Arrows point to the defect region. Scale bar, 1 mm. (**D**) Bone volume fraction of intramedullary and cortical defect regions at days 7 and 21 post-injury. *n* = 4–6 mice/group. Data are expressed as means ± SD and analyzed by unpaired two-tailed t-test.

The online version of this article includes the following source data for figure 7:

**Source data 1.** Source data for *Figure 7B*.

**Source data 2.** Source data for *Figure 7D*.

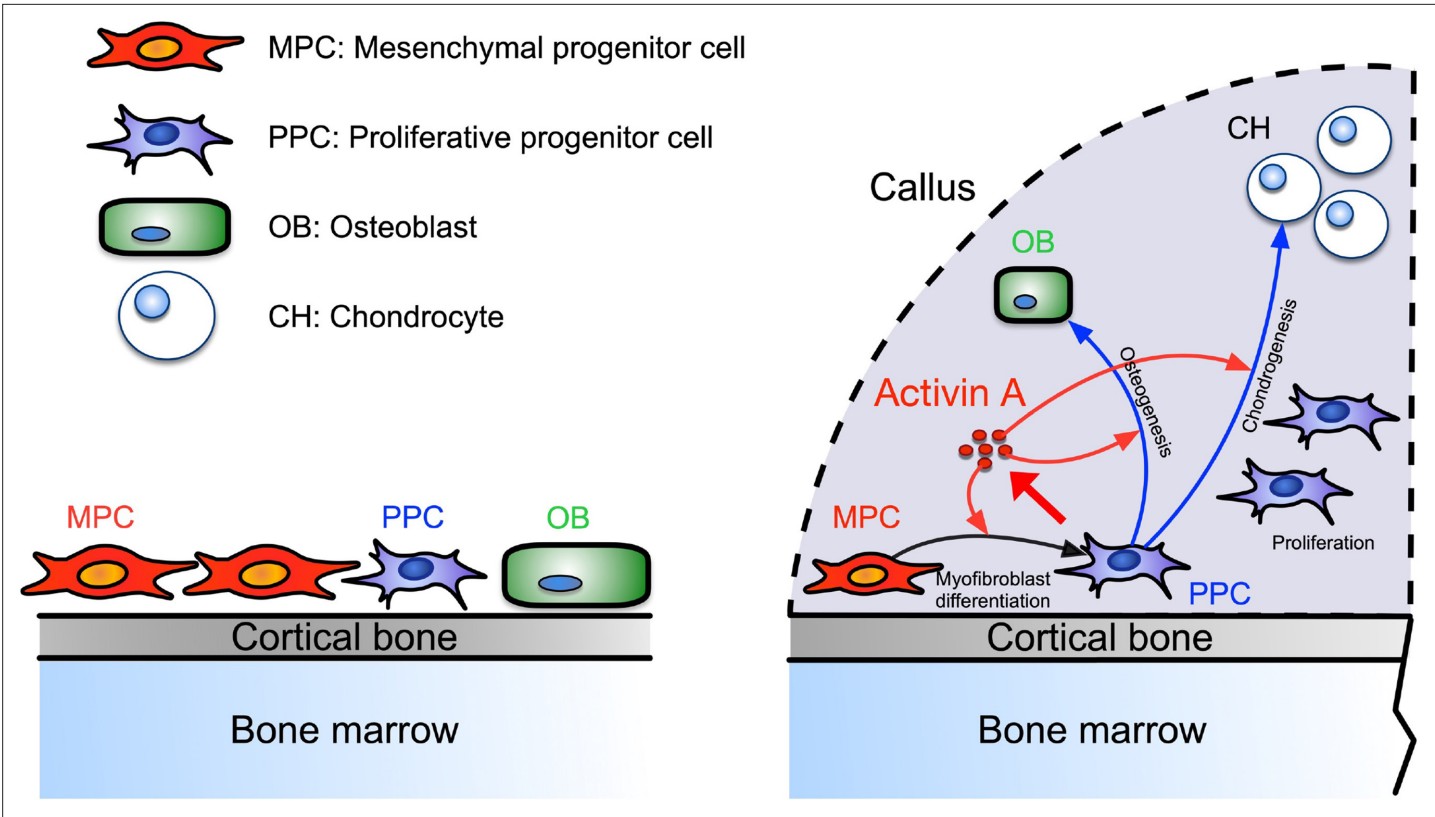

**Figure 8.** Working model of Activin A roles in callus during progression of fracture healing.

The online version of this article includes the following figure supplement(s) for figure 8:

**Figure supplement 1.** Markers of Pi16+ cells from a recent single-cell RNA-sequencing (scRNA-seq) study of fibroblasts are specifically expressed in the mesenchymal progenitor cell (MPC) cluster.

and based on μCT-based sagittal and cross-sectional reconstitution, volume fraction of reconstituted bone (BV/TV) was over 60% (*Figure 7B*). Systemic nActA.AB administration caused an appreciable reduction in bone deposition in both drilled and medullary regions by day 7 and a significant drop by day 21 compared to isotype controls (*Figure 7A, B*). Conversely, local supplementation of recombinant Activin A significantly increased BV/TV in medullary and drilled regions at both days 7 and 21 compared to vehicle controls (*Figure 7C, D*). Our data indicate that Activin A also promotes bone regeneration via intramembranous ossification.

## Discussion

Our data identify a novel population of PPCs that rapidly expands within the developing callus after fracture, is characterized by high Activin A/Inhba expression, and displays a myofibroblast-like character. Based on scRNA-seq-based trajectory analysis, the PPCs lie at the center of a developmental path that bifurcates and elicits the emergence of chondrocytes and osteoblasts within the callus over time (see schematic in *Figure 8*). Activin A expression and function in these and other cells appear to be critical for effective bone healing given that repair in both our endochondral and intramembranous mouse models was significantly delayed by systemic administration of Activin A neutralizing antibody. This key notion is reinforced by our findings that the same healing processes were boosted by local supplementation of recombinant Activin A. Indeed, the exogenous protein robustly enhanced pSMAD2 signaling levels within the fracture callus and promoted the acquisition of a myofibroblast-like phenotype by the progenitor cells and their subsequent chondrogenic and osteogenic differentiation in vitro. Overall, our data and insights are very much in line with a previous study in rats in which local implantation of recombinant Activin A stimulated fracture repair (*Sakai et al., 1999*). They also agree quite well with siRNA studies showing that endogenous Activin A is required for chondrogenic

and osteogenic differentiation of human marrow mesenchymal stem cells (*Djouad et al., 2010*). However, we should mention fracture studies employing soluble neutralizing type IIA or IIB receptors that elicited different outcomes, possibly due to the ability of soluble receptors to interfere with not only Activin A but other superfamily members also (*Pearsall et al., 2008*; *Puolakkainen et al., 2017*). In sum, our study provides clear evidence that Activin A is an overall regulator and stimulator of the fracture repair process. The protein appears to act by promoting myofibroblastic, chondrogenic and osteogenic differentiation and ultimately bone healing, with the PPC population placed at the center of a developmental cascade aiding the progression of the overall process. Our data also establish Activin A as a potential therapeutic tool to enhance fracture repair.

Periosteal mesenchymal stem and progenitor cells are well known to be essential for fracture repair, but the sequential steps needed to turn them into reparative cells and mechanisms underlying this multifaceted phenotypic progression have remained unclear (*Einhorn and Gerstenfeld, 2015*; *Loi et al., 2016*). The distinct PPC population identified here occupies an intermediate developmental step along the repair cascade and possesses a myofibroblast-like character after injury (*Hinz, 2016*). Prior to fracture injury, the periosteum contains few PPCs, but fracture triggers a concerted response in which MPCs are rapidly activated to become PPCs, differentiating into chondrocytes and osteoblasts with time. Based on cell prevalence, the PPCs appear to represent the bulk of cells within the thickening periosteum a few days after fracture. Notably also, the PPCs express not only *Inhba* but also *Acta2* that encodes αSMA, a well-established marker of periosteal mesenchymal progenitors engaged in fracture callus development (*Grcevic et al., 2012*). Using *Acta2-CreER* mice, the Kalajzic group recently demonstrated that αSMA+ cells constitute about 4% of Cd45−Ter119−Cd31− cells in homeostatic mouse periosteum and that DTA-mediated ablation of the cells severely reduces callus size after fracture (*Matthews et al., 2021*). In the tissue injury and regeneration field, αSMA is often used as a marker of myofibroblasts, a cell population first discovered in skin wound healing studies and then identified as a key player in many tissue repair processes (*Hinz, 2016*; *Pakshir et al., 2020*). A recent scRNA-seq study isolated fibroblast cell populations from 13 injured and diseased mouse tissues including bone, and identified an Lrrc15+ cell cluster that displays a αSMA+ myofibroblastic cell character and emerges only after injury (*Buechler et al., 2021*). Interestingly, Lrrc15 is also a marker for PPCs in our study. In addition, abundant αSMA+ stromal cells were recently reported to occur around the injury site after metal implant surgery in mouse tibiae (*Vesprey et al., 2021*). In sum, acute injury resulting from bone fracture, metal implantation, or other insults appears to elicit a common and forceful repair response that is coupled to the emergence of progenitors with a myofibroblast-like phenotype. Ongoing studies are directed toward deciphering more precisely what roles PPCs perform in bone repair and what mechanisms ensure the controlled regression of the cells and the apparent loss of myofibroblastic-like features at later stages of facture healing.

Extensive studies have been performed previously to identify periosteal mesenchymal stem and progenitor cells that generate the callus in response to bone fracture. Those studies generally made use of one or a combination of markers to define progenitors by flow cytometry or lineage tracing approaches. Using an unbiased, comprehensive scRNA-seq approach that covers the entire mesenchymal lineage cell populations in periosteum, our study computationally identifies MPCs as the most primitive form of progenitors that give rise to other mesenchymal populations in both intact and fractured bones. Our data indicate also that most of the previously identified progenitor cell markers including Cd200, Cd105 (Eng), and Cd51 (Itgav) are broadly expressed amongst periosteal mesenchymal populations (*Chan et al., 2015*; *Debnath et al., 2018*). These MPCs appear to be quite similar to those in a recent study indicating that sorted Sca1+ and Cd90+ periosteal cells have high CFU-F potential and multi-differentiation abilities (*Matthews et al., 2021*). They also share a stem cell gene profile including *Ly6a*, *Thy1*, *Cd34*, and *Clec3b* that is similar to that of early mesenchymal progenitors previously identified in mouse bone marrow mesenchymal cells (*Zhong et al., 2020*). The recent global scRNA-seq study analyzing fibroblasts, defined as Pdgfrα+ cells, from 16 mouse tissues cited above showed that fibroblasts in all tissues, regardless of their states (steady and perturbed), contain a most primitive cell cluster termed Pi16+ cells (*Buechler et al., 2021*). Notably, markers of Pi16+ cells (*Pi16*, *Dpp4*, *Ly6c1*, and *Dpt*) are also MPC markers (*Figure 8—figure supplement 1*), and MPC markers such as Ly6a and Cd34 are also highly and specifically expressed in Pi16+ cells (*Buechler et al., 2021*). Together, the above data and insights highlight the shared characteristics of fibroblast-like stem cells at different anatomic sites. It will be interesting to create CreER lines to trace MPCs

in vivo and validate their primitive position relative to differentiation trajectories and developmental fate.

It has long been appreciated that Activin A is produced by inflammatory cells (*de Kretser et al., 2012*; *Morianos et al., 2019*). These cell populations are present at the bone repair site and are found to be essential for effective fracture repair (*Loi et al., 2016*; *Mountziaris et al., 2011*; *Kolar et al., 2010*). Inflammatory cells present at the fracture repair site include neutrophils and macrophages and release a spectrum of inflammatory and chemotactic mediators including members of the TNF and IL protein families (*Einhorn and Gerstenfeld, 2015*; *Loi et al., 2016*). These and other proteins are thought to lead to recruitment of fibroblasts, MSCs, and skeletogenic progenitors from local sources, propelling the next phase of skeletal tissue repair production and deposition (*Gerstenfeld et al., 2001*; *Gerstenfeld et al., 2003*), but details remain scant (*Loi et al., 2016*). The previously described roles of Activin A in inflammation (*de Kretser et al., 2012*) suggest that the protein may promote the initial inflammatory response needed for setting the repair process. Our data here and in a previous report from our labs show that Activin A is produced by skeletogenic cells and promotes bone formation (*Mundy et al., 2021*). The present study builds on those findings and reveals more critically that *Inhba* becomes prominently expressed by the myofibroblast-like PPCs within the callus and that Activin A promotes chondrogenesis and osteogenesis in periosteal cells. The protein is also a product of chondrocytes within the healing callus. Together, previous studies and our current data lead to the important notion that Activin A could represent a regulatory and functional nexus linking inflammation to local skeletogenic responses in diverse cell populations and in turn, boosting fracture repair progression.

Activin A's capacity to play such diverse biological roles could endow the protein with ideal characteristics as a therapeutic for fracture repair deficiencies. A variety of means have been tested in both animal studies and patients to improve fracture healing, but an effective and safe therapy is yet to emerge and be clinically applied (*Einhorn and Gerstenfeld, 2015*; *Roberts and Ke, 2018*). Anabolic therapies utilize exogenous agents such as members of the BMP, FGF, Wnt, or hedgehog protein families and may not be as effective as desirable because of their targeting a given step or a given population mainly (*Roberts and Ke, 2018*). Activin A may prove more effective because of its early presence from the very onset of the fracture repair process and its activity through sequential populations, including our newly described myofibroblast-like node toward terminal differentiation of cartilage and bone cells. These paradigms remain to be fully tested in the future, particularly using genetic approaches in which *Inhba* could be conditionally ablated at different stages and different populations to delineate its function in each. Similarly, our loss of function data using the Activin A neutralizing antibody could be influenced by systemic action away from the fracture site. These are limitations of the current study that will need to be tackled directly in future studies. Nonetheless, the spectrum of biological and physiologic action by Activin A and the appreciable promotion of fracture repair we demonstrate here provide a rational foundation and premise for the investigation of Activin A for clinical application.

## Methods

**Key resources table**

| Reagent type (species) or resource | Designation | Source or reference | Identifiers | Additional information |
|---|---|---|---|---|
| Genetic reagent (*Mus musculus*) | Col2a1-Cre | Jackson Laboratory | Stock #: 003554 | |
| Genetic reagent (*Mus musculus*) | Rosa26^LSL-tdTomato | Jackson Laboratory | Stock #: 007909 | |
| Genetic reagent (*Mus musculus*) | C57BL/6 | Jackson Laboratory | Stock #: 000664 | |
| Antibody | Mouse monoclonal neutralizing antibody against Activin A | Biolegend | Cat #: 693604 | 10 mg/kg |
| Antibody | Rabbit monoclonal anti-mouse pSMAD2 | Cell Signaling | Cat #: 3108S | 1:200 |
| Antibody | Mouse monoclonal anti-mouse αSMA | Sigma | Cat #: A5228 | 1:200 |

*Continued on next page*

*Continued*

| Reagent type (species) or resource | Designation | Source or reference | Identifiers | Additional information |
|---|---|---|---|---|
| Antibody | Goat polyclonal anti-mouse Activin A | R&D Systems | Cat #: AF338 | 1:200 |
| Antibody | Alexa Fluor 647 donkey polyclonal anti-goat | Invitrogen | Cat #: A-21447 | 1:200 |
| Antibody | Alexa Fluor 488 donkey polyclonal anti-rabbit | Invitrogen | Cat #: A-21206 | 1:200 |
| Antibody | Alexa Fluor 488 donkey polyclonal anti-mouse | Invitrogen | Cat #: A-21202 | 1:200 |
| Antibody | Alexa Fluor 555 donkey polyclonal anti-mouse | Invitrogen | Cat #: A-31570 | 1:200 |
| Antibody | Rat monoclonal anti-mouse Ter119 FITC | Biolegend | Cat #: A-116205 | 1:100 |
| Antibody | Rat monoclonal anti-mouse CD31 FITC | Biolegend | Cat #: A-102509 | 1:100 |
| Antibody | Rat monoclonal anti-mouse CD45 FITC | Biolegend | Cat #: A-147709 | 1:100 |
| Antibody | Rat monoclonal anti-mouse CD34 BV421 | BD Biosciences | Cat #: A-562608 | 1:100 |
| Commercial kit | Click-iT Plus EdU Alexa Fluor 647 Flow Cytometry Assay Kit | Thermo Fisher | Cat #: A-C10424 | |
| Peptide, recombinant protein | Recombinant Activin A | R&D Systems | Cat #: 338-AC-010 | 10 µg |
| Software | Cellranger | https://support.10xgenomics.com | Version 6.0.1 | |
| Software | ImageJ software | ImageJ (http://imagej.nih.gov/ij/) | | |
| Software | GraphPad Prism software | GraphPad Prism (https://graphpad.com) | | |

## Animal models

Specific pathogen-free 2- and 20-month-old C57BL/6 female mice were purchased from the Jackson Laboratory (000664) for treatment studies. Col2a1-Cre Rosa26*LSL-tdTomato* (Col2/Td) mice were generated by breeding Rosa26*LSL-tdTomato* (Jackson Laboratory, 007909) mice with Col2a1-Cre (Jackson Laboratory, 003554) (*Ovchinnikov et al., 2000*) mice. Since we did not detect any gender difference in fracture healing using these mice, a mixture of male and female 2-month-old mice were used for periosteal cell isolation, scRNA-seq, and histological analyses. For mice receiving a fracture, closed transverse fractures were made on right tibiae via a blunt guillotine with a pre-inserted intramedullary pin for stabilization as previously described (*Wang et al., 2019*). For mice receiving a drill hole, a 0.8-mm diameter unicortical drill-hole defect was made using a drill bit first followed by a 21-G needle in the diaphysis part of right femur (*Li and Helms, 2021*). For systemic antibody treatment, mouse monoclonal neutralizing antibody against Activin A (nActA.AB, Biolegend, 693604) or pre-immune control IgG2b isotype (Biolegend, 400377) were subcutaneously injected twice per week (10 mg/kg) after fracture. This antibody was utilized in our previous study on HO in which we showed that it is specific for Activin A and does not interfere with Activin B (*Mundy et al., 2021*). For recombinant Activin A implantation treatment, the protein (R&D Systems, 338-AC-010, 1 µg) was mixed with growth factor-reduced, phenol red-free Matrigel (Corning, 356231) per 50 µl total aliquot volume. Tibial anterior side was minimally exposed immediately after fracture and the Matrigel mix was applied at the fracture site using an insulin syringe. The skin was then closed with sutures.

## Human fracture samples and intact iliac periosteum samples

Fracture tissue sections were prepared from the de-identified surgical discard specimens obtained from the open reduction and internal fixation surgeries of patients at days 3–10 post long bone fractures

(*n* = 12). Human intact iliac periosteum samples were prepared from the de-identified surgical discard specimens obtained from patients undergoing autologous bone grafting surgery (*n* = 9).

## Periosteum Td⁺ cell isolation and cell sorting

Periosteum cells were harvested as described previously (*Wang et al., 2019*). At day 0 before fracture and day 5 after fracture, tibiae were dissected free of surrounding tissues and both epiphyseal ends were sealed with 3% agarose gel. The remaining bone fragments were digested in 2 mg/ml collagenase A and 2.5 mg/ml trypsin. Cells from the first 3 min of digestion were discarded and cells from a subsequent 30 min of digestion were collected as periosteal cells. For samples harvested on day 10 after fracture, the tibiae were dissected free of surrounding tissues, and fracture calluses were cut off using a surgical blade and cut into small pieces. The fragments were digested in 2 mg/ml collagenase A and 2.5 mg/ml trypsin for 1 hr and collected as whole callus cells. For sorting, cells were resuspended into fluorescence-activated cell sorting (FACS) buffer containing 25 mM 4-(2-Hydroxyethyl)-1-piperazine ethanesulfonic acid (HEPES; Thermo Fisher Scientific) and 2% fetal bovine serum (FBS) in phosphate-buffered saline (PBS) and sorted for Td⁺ cells using Influx B or Aria B (BD Biosciences).

## scRNA-seq of periosteal mesenchymal cells

We constructed three batches of single-cell libraries for sequencing using sorted periosteum Td⁺ cells from day 0 before fracture (*n* = 5 mice, 3 males and 2 females), day 5 after fracture (*n* = 6 mice, 4 males and 2 females), and day 10 after fracture (*n* = 6 mice, 3 males and 3 females). Approximately 20,000 cells were loaded each time into Chromium controller (V3 chemistry version, 10X Genomics Inc), barcoded and purified as described by the manufacturer, and sequenced using a 2 × 150 pair-end configuration on an Illumina Novaseq platform at a sequencing depth of ~400 million reads. Cell ranger (Version 6.0.1, https://support.10xgenomics.com/single-cell-geneexpression/software/pipelines/latest/what-is-cell-ranger) was used to demultiplex reads, followed by extraction of cell barcode and UMIs. The cDNA insert was aligned to a modified reference mouse genome (mm10).

Seurat package V3 (*Stuart et al., 2019*) was used for individual or integrated analysis of the datasets. Standard Seurat pipeline was used for filtering, variable gene selection, dimensionality reduction analysis, and clustering. Doublets or cells with poor quality (genes >7000, genes <1000, or >10% genes mapping to mitochondrial genome) were excluded. Expression was natural log transformed and normalized for scaling the sequencing depth to a total of $1 \times 10^4$ molecules per cell. Seurat cell cycle scoring function were used to analyze cell proliferation, proliferative cells were defined as cells in G2M or S phase. First identify the top 2000 variable genes by controlling for the relationship between average expression and dispersion. Then, expression matrix were scaled by regressing out cell cycle scores (G2M.Score and S.Score). Statistically significant principal components (PC) were selected as input for UMAP plots. For the integrated dataset, batch integration was performed using Harmony (version 1.0) (*Korsunsky et al., 2019*). Different resolutions for clustering were used to demonstrate the robustness of clusters. In addition, differentially expressed genes within each cluster relative to the remaining clusters were identified using FindMarkers. Sub-clustering was performed by isolating the mesenchymal lineage clusters using known marker genes, followed by reanalysis as described above. Mesenchymal lineage cells, excluding tenocytes and synovial fibroblasts, were selected for sub-clustering and reanalysis. Gene ontology analysis was performed using the clusterProfiler package (*Yu et al., 2012*).

To computationally delineate the developmental progression of periosteal mesenchymal cells and order them in pseudotime, we performed the trajectory analysis using Slingshot (*Street et al., 2018*). To do so, Seurat objects were transformed into SingleCellExperiment objects. Slingshot trajectory analysis was conducted using the Seurat clustering information and with dimensionality reduction produced by UMAP.

RNA velocity analysis was performed as described (*La Manno et al., 2018*). Briefly, Velocyto was used to generate count tables for spliced and unspliced transcripts that were then processed through the aforementioned Seurat pipeline to produce UMAP projection and clustering information. All above was then input into Scvelo for visualizing directed RNA dynamic information using dynamical model (*Bergen et al., 2020*).

## µCT analysis

Tibiae harvested post fracture were scanned at the fracture sites by VivaCT 40 (Scanco Medical AG) at a 7.4-µm isotropic voxel size to acquire a total of 1000 µCT slices centering around the fracture site. A semi-automated contouring method was used to determine the callus perimeter and to analyze the callus outside the preexisting cortical bone. All images were first smoothed by a Gaussian filter (sigma = 1.2, support = 2.0) and then applied by a threshold corresponding to 30% of the maximum available range of image gray scale values to distinguish mineralized tissue from unmineralized and poorly mineralized tissue. Callus region surrounding cortical bone was contoured for trabecular bone analysis. Based on µCT images, 6 weeks fracture samples were assigned fracture healing scores according to an 8-point radiographic scoring system (*An and Friedman, 1999*). This is a sum of scores from three categories: periosteal reaction (0–3), bone union (0–3), and remodeling (0–2). Scoring was determined empirically by two independent experts who were blinded to treatment allocation. To analyze bone healing after drill-hole injury, the cortical defect area and the intramedullary area were contoured separately for trabecular bone analysis.

## Mechanical testing

Tibiae at 6 weeks post fracture were harvested for mechanical testing using an Instron 5542 (Instron, Norwood, MA, USA) as described previously (*Wang et al., 2019*). Tibiae were positioned so that the loading point was at the fracture site. A load speed of 1.8 mm/min was applied midway between two supports placed 10 mm apart. Peak load, stiffness, and energy to failure were calculated from the force-to-failure curve.

## Histology and immunohistochemistry

Fractured mouse tibiae were fixed in 4% Paraformaldehyde (PFA), decalcified in 10% Ethylenediaminetetraacetic acid (EDTA) for 3 weeks, and processed for paraffin embedding. A series of 6-µm-thick longitudinal sections were cut across the entire fracture callus from one side of cortical bone to the other side of cortical bone. For each bone, a central section with the largest callus area as well as two sections at 192 µm (~1/4 bone width) before and after the central section were stained with Safranin-O/Fast green and quantified for cartilage area, bone area, and fibrosis area by ImageJ. Additional sections neighboring the central section were used for IHC. After antigen retrieval, slides were incubated with rabbit anti-pSMAD2 (S465/467) (Cell Signaling, 3108S) and mouse anti-αSMA (Sigma, A5228) primary antibodies at 4°C overnight, followed by incubation with Alexa Fluor 488 donkey anti-rabbit (Invitrogen, A-21206) and Alexa Fluor 555 donkey anti-mouse (Invitrogen, A-31570) secondary antibodies for 1 hr at RT. Sections were scanned by a Nikon Eclipse 90i fluorescence microscope. To quantify positive cells, 4 square regions (0.25 mm$^2$ each) evenly distributed in the thickened periosteum were selected at similar locations in each sample. Within these regions, pSMAD2+ or αSMA+ cells were counted and normalized against total 4',6-diamidino-2-phenylindole (DAPI)+ cells.

To obtain whole mount sections for immunofluorescent imaging, freshly dissected mouse bones were fixed in 4% PFA for 1 day, decalcified in 10% EDTA for 4–5 days, and immersed into 20% sucrose and 2% polyvinylpyrrolidone (PVP) at 4°C overnight. Samples were embedded in embedding medium containing 8% gelatin, 20% sucrose, and 2% PVP and sectioned at 50 µm in thickness. Sections were incubated with mouse anti-αSMA, goat anti-Activin A (R&D, AF338) primary antibodies at 4°C overnight, followed by incubation with Alexa Fluor 488 donkey anti-mouse (Invitrogen, A-21202) and Alexa Fluor 647 donkey anti-goat (Invitrogen, A-21447) secondary antibodies for 1 hr at RT. Fluorescence images were captured by a Zeiss LSM 710 scanning confocal microscope interfaced with the Zen 2012 software (Carl Zeiss Microimaging LLC).

After collection, human samples were fixed in 4% PFA overnight, followed by paraffin embedding and staining with H&E. For IHC of Activin A, slides were incubated with anti-human Activin A antibody (R&D, AF338) at 4°C overnight, followed by binding with biotinylated secondary antibody and DAB color development.

## Flow cytometry analysis of EdU+ cells

Mice received 1.6 mg/kg EdU at 3 hr before sacrifice. Digested periosteal cells were stained with rat anti-Ter119 FITC (Biolegend, 116205), rat anti-CD31 FITC (Biolegend, 102509), rat anti-CD45 FITC (Biolegend, 147709), and rat anti-CD34 BV421 (BD Biosciences, 562608). EdU detection was carried

out according to the manufacturer's instructions (Click-iT Plus EdU Alexa Fluor 647 Flow Cytometry Assay Kit, Thermo Fisher Scientific, C10424). Flow cytometry was performed by either LSR A or BD LSR Fortessa flow cytometer and analyzed by FlowJo v10.5.3 for MAC.

## Cell culture

Enzymatically released mouse periosteal cells were seeded in the growth medium (αMEM supplemented with 15% FBS plus 55 μM β-mercaptoethanol, 2 mM glutamine, 100 IU/ml penicillin, and 100 μg/ml streptomycin) for periosteal mesenchymal progenitor culture. For CFU-F assay, unsorted cells, sorted Td+ cells, and sorted Td− cells were seeded at $1 \times 10^6$, $3 \times 10^4$, and $1 \times 10^6$ cells per T25 flask, respectively. Seven days later, flasks were stained with 3% crystal violet to quantify CFU-F numbers. For cell proliferation assay, 1000 cells were seeded into a 96-well plate in the growth medium containing Activin A (100 ng/ml) or nActA.AB (100 μg/ml). Cell numbers at days 0, 1, 2, and 3 were quantified using CyQUANT Proliferation Assay Kit (Invitrogen, C35011). For myofibroblast differentiation assay, $0.2 \times 10^6$ cells were seeded into a 6-well plate with serum-free medium containing Activin A (100 ng/ml) or TGF-β1 (10 ng/ml) for 72 hr. Cells were then stained with mouse anti-αSMA primary antibody for 1 hr followed by Alexa Fluor 488 donkey anti-mouse secondary antibody for 1 hr. For osteogenic differentiation assay, confluent cells were switched to osteogenic medium (αMEM with 10% FBS, 10 nM dexamethasone, 10 mM β-glycerophosphate, 50 μg/ml ascorbic acid, 100 IU/ml penicillin, and 100 μg/ml streptomycin) containing Activin A (100 ng/ml) or nActA.AB (100 μg/ml) for 2 weeks followed by alizarin staining. For chondrogenic differentiation, micromass cultures were initiated by spotting 20 μl of cell suspension ($0.5 \times 10^6$ cells/spot) onto the surface of a 24-well plate. After 2 hr incubation at 37°C in a humidified $CO_2$ incubator to allow for cell attachment, the cultures were switched to basic chondrogenic medium (high glucose Dulbecco's modified Eagle medium, 100 μg/ml sodium pyruvate, 1% insulin, human transferrin, and selenous acid (ITS)+ Premix, 50 μg/ml ascorbate-2-phosphate, 40 μg/ml L-proline, 0.1 mM dexamethasone, 100 IU/ml penicillin, and 100 μg/ml streptomycin) containing Activin A (100 ng/ml) or nActA.AB (100 μg/ml) for 2 weeks followed by alcian blue staining.

## Quantitative RT-PCR analysis

Fracture callus was dissected out from human or mouse tissues and snap frozen in liquid nitrogen and minced into powder. RNA was extracted by adding TRIzol Reagent to the powder and further purified by RNeasy Micro Kit (QIAGEN, 74004). Sorted or cultured cells were collected in TRIzol Reagent for RNA purification (Sigma, T9424). A High-Capacity cDNA Reverse Transcription Kit (Thermo Fisher Scientific, 4368814) was used to reverse transcribe mRNA into cDNA. Following this, real-time PCR was performed using a Power SYBR Green PCR Master Mix Kit (Thermo Fisher Scientific, Inc, 4367659). The primer sequences for genes used in this study are listed in *Supplementary file 1c*. The gene expression level was normalized against the internal housekeeping gene *Actb*.

## Statistical methodologies

Data are expressed as means ± standard deviation and analyzed by *t*-tests or one-way analysis of variance with Tukey post-test for multiple comparisons using Prism software (GraphPad Software, San Diego, CA). For cell culture experiments, observations were repeated independently at least three times with a similar conclusion, and only data from a representative experiment are presented. Values of p < 0.05 were considered significant.

## Study approval

The experimental animal protocols were approved by the Institutional Animal Care and Use Committees of the University of Pennsylvania (IACUC# 804112) and the Children's Hospital of Philadelphia (IACUC# 20-000958). The experiments were performed in the animal facilities of both institutions, which implement strict regimens for animal care and use. In accordance with the standards for animal housing, mice were group housed at 23–25°C with a 12-hr light/dark cycle and allowed free access to water and standard laboratory pellets.

## Acknowledgements

We thank Dr. Ivo Kalajzic at the University of Connecticut Health Center for critical reading of the manuscript and suggestions, and the Penn Center for Musculoskeletal Disorders (PCMD) at the University of Pennsylvania for the expert use of its histology, µCT imaging, and biomechanics core facilities. This study was supported by National Institutes of Health grants R01AR071946 (MP), R21AR074570, R01AG069401 (LQ), K99AR078352 (DR) and P30AR069619 (PCMD).

## Additional information

### Funding

| Funder | Grant reference number | Author |
|---|---|---|
| National Institutes of Health | R01AR071946 | Maurizio Pacifici |
| National Institutes of Health | R21AR074570 | Ling Qin |
| National Institutes of Health | R01AG069401 | Ling Qin |
| National Institutes of Health | K99AR078352 | Danielle Rux |

The funders had no role in study design, data collection, and interpretation, or the decision to submit the work for publication.

### Author contributions

Lutian Yao, Conceptualization, Data curation, Software, Formal analysis, Investigation, Methodology, Writing – original draft, Writing – review and editing; Jiawei Lu, Investigation, Methodology; Leilei Zhong, Formal analysis, Investigation; Yulong Wei, Tao Gui, Luqiang Wang, Methodology; Jaimo Ahn, Joel D Boerckel, Danielle Rux, Conceptualization, Methodology; Christina Mundy, Conceptualization, Data curation, Investigation, Methodology; Ling Qin, Conceptualization, Supervision, Funding acquisition, Writing – original draft, Writing – review and editing; Maurizio Pacifici, Conceptualization, Data curation, Supervision, Funding acquisition, Writing – original draft, Writing – review and editing

### Author ORCIDs

Lutian Yao https://orcid.org/0000-0002-0652-2075
Leilei Zhong http://orcid.org/0000-0003-1153-4115
Yulong Wei http://orcid.org/0000-0003-3823-9984
Joel D Boerckel http://orcid.org/0000-0003-3126-3025
Christina Mundy http://orcid.org/0000-0003-3328-5965
Ling Qin http://orcid.org/0000-0002-2582-0078
Maurizio Pacifici http://orcid.org/0000-0001-6854-4942

### Ethics

The experimental animal protocols were approved by the Institutional Animal Care and Use Committees of the University of Pennsylvania (IACUC#804112) and the Children's Hospital of Philadelphia (IACUC#20-000958). The experiments were performed in the animal facilities of both institutions, which implement strict regimens for animal care and use.

### Decision letter and Author response

Decision letter https://doi.org/10.7554/eLife.89822.sa1
Author response https://doi.org/10.7554/eLife.89822.sa2

## Additional files

### Supplementary files

• Supplementary file 1. Supplementary tables. (**a**) Cell numbers and percentages are listed for cell clusters at day 0 before fracture and days 5 and 10 after fracture. (**b**) Cell numbers and percentages

are listed for cell clusters of periosteal mesenchymal lineage cells at day 0 before fracture or days 5 and 10 after fracture. (**c**) Mouse real-time RT-PCR primer sequences used in this study.

• MDAR checklist

### Data availability

All data needed to evaluate the conclusions of this study are present in the paper and/or supplementary material. Sequencing data have been deposited in GEO under accession code GSE192630.

The following dataset was generated:

| Author(s) | Year | Dataset title | Dataset URL | Database and Identifier |
| --- | --- | --- | --- | --- |
| Yao L, Qin L, Pacifici M | 2024 | Activin A promotes mouse bone fracture repair and characterizes a novel myofibroblastic population in callus | https://www.ncbi.nlm.nih.gov/geo/query/acc.cgi?acc=GSE192630 | NCBI Gene Expression Omnibus, GSE192630 |

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
