## [Editor Report]

This important work identified a novel role for Activin A in promoting long bone fracture repair while also demonstrating its therapeutic potential. The evidence supporting the conclusion that Activin A is an important inducer of chondrocyte and osteoblast differentiation that contributes to bone healing is convincing. This work describes novel and valuable findings that will be of interest to both scientists and clinicians in the musculoskeletal field.

---

## [Decision Letter]

**Decision letter after peer review:**

Thank you for submitting your article "Activin A marks a novel progenitor cell population during fracture healing and reveals a therapeutic strategy" for consideration by *eLife*. Your article has been reviewed by 3 peer reviewers, and the evaluation has been overseen by a Reviewing Editor and Hiroshi Takayanagi as the Senior Editor. The following individual involved in the review of your submission has agreed to reveal their identity: Ugur Ayturk (Reviewer #3).

Essential revisions:

This is a generally well-executed set of study, the data from which are appropriate and largely supports the conclusions. The manuscript would be strengthened by convincingly demonstrating that the PPCs are the primary mediator of Activin A actions during fracture repair.

*Reviewer #1 (Recommendations for the authors):*

Questions and concerns:

1. Multiple pieces of data show Inhba/ACTIVIN A expression in a number of other cell types, but a majority of the language pins all of the responses to ACTIVIN A as coming from the PPCs and disregarding the other non-mesenchymal cell types. For example, in Sup. Figure 1 Inhba expression is quite high in the granulocyte clusters (13, 14, 15). While PPC produced ACTIVIN A is likely a strong participant, further knock-out data in the PPCs and other cell types of would be required to claim this. The final paragraph of the discussion includes this, but the language should be toned down throughout the results. If the language is to be kept, further cell specific KO data is required utilizing the Acta2 specific Cre KO of Inhba.

2. Please ensure the proper formatting for genes/proteins is correct throughout the manuscript (ie when referring to the protein ACTIVIN A should be all caps).

3. With the neutralizing antibody usage, how do you know it is specifically targeting the mesenchymal cells (page 13 top paragraph)? Only giving percentages of cells stained for pSMAD2 and aSMA does not explicitly demonstrate that these PPCs are the responsible party in decreasing fracture healing parameters. Co-staining as well as assessing potential decrease in pSMAD2 staining in other cell types may shed light on ACTIVIN A functions in other cell types.

4. What is the expression pattern of the ACTIVIN A receptor? This may allow for further conclusions. Additionally, in the in vitro MPC sorted rACTIVIN A experiments, markers of myofibroblast emerge after treatment and in the chondrogenic/osteogenic media there is further differentiation upon treatment. Are the authors suggesting there is a self-feedback mechanism on the PPCs from their own ACTIVIN A expression?

5. Are cells in the bone only responsive to ACTIVIN A signaling after injury, or could there be effects of the protein in an uninjured setting?

*Reviewer #2 (Recommendations for the authors):*

In this study, Lutian Yao et al. have found that TGF-b family member Activin A is expressed in proliferative progenitor cells (PPCs) which was identified in the tibia by single cell analysis. They also demonstrated that PPCs gain myofibroblasts feature after fracture. They found that Activin A can stimulates proliferation and differentiation of periosteal progenitors and promote chondrogenesis and osteogenesis during fracture healing process. The authors have combined the fracture model with single cell analysis to investigate the function of Activin A during the bone healing process. Overall, the discovery from this study has important implications for the bone healing process during fracture. However, several important questions remain to be clarified in this manuscript. Moreover, the mechanistic insight of this manuscript still needs to be enhanced.

1. In Figure 2D, the authors only showed the cell cycle stage for the populations they identified. The expression of cell proliferation marker also needs to be provided to further prove the proliferative feature of PPCs.

2. In Figure 2F and supplemental figure 6, the velocity analysis did not clearly show the trajectory of EOB. The authors should explain this.

3. In Figure 3F, the staining of Activin A is confusing. Which color represents Activin A?

4. The authors have shown that Activin A can stimulate the proliferation and differentiation of periosteal progenitors in Figure 4. What is the mechanism behind this?

5. The authors have demonstrated that Activin A can accelerate the fracture healing process. What is possible role of TGFb signaling in this process?

*Reviewer #3 (Recommendations for the authors):*

Below are some specific comments and questions for the authors:

– It is unclear from Figure 1A if mesenchymal progenitors in bone marrow express Activin A. Based on the single cell RNA-seq data, it is possible that the majority of labeled marrow cells are granulocytes. This can be checked with higher magnification imaging of slides co-stained with CD31 (as the progenitors will likely be pericytes) or better yet LEPR.

– "Since almost all cells in the thickened periosteum at day 5 are Td+, this observation suggests that muscle-derived cells do not significantly contribute to the early stage of fracture healing". Not sure if your data supports this conclusion. Whatever portion of muscle cells participate in fracture healing, they may not express Col2-cre at baseline but start expressing it shortly after they are mobilized and differentiating, making it difficult to distinguish them from cells with periosteal or marrow origin in your system. Also, fibroadipogenic progenitors (if they are Td+) can be difficult to spot at the magnification the data are presented. Have you checked intact muscle tissue with FACS for Td+ cells?

– I am not sure if it is possible to call your cell collections exclusively "periosteal", given the anatomically broad recombination profile. The Methods sections suggests that the intact bones were not crushed, and it seems that the cell clusters do not include LEPR+/CAR cells, which are both reassuring. However, such control is difficult (or perhaps not even possible) when dealing with developing fracture callus.

– "Particularly relevant to the present study was the finding that compared to MPCs, PPCs highly expressed Inhba after fracture (Figure 3C and Supplementary Figure 7B). Note that chondrocytes also highly expressed Inhba, consistent with the immunostaining results shown in Figure 1. However, their number was much lower than PPCs in early callus (Supplementary Figure 4B), suggesting that PPCs are the main source of activin A in early fracture healing." The violin plot in Figure 3C alone does not support this statement. There appears to be a remarkable increase in Inhba expression in chondrocytes (which is consistent with Figure 1A-B), but less so in other cells. Further, SuppFig4B presents an estimate of cell counts, which might very well be biased by the efficiency of the flow cytometer and/or 10X platform, when comparing fibroblast-like cells and chondrocytes.

– "Based on scRNA-seq data, we sorted Cd45-Cd31- Ter119-Cd34+ (Lin-/Cd34+) cells and Cd45-Cd31-Ter119-Cd34- (Lin-/Cd34-) cells to represent MPCs and PPCs, respectively." Cluster-specific quantification of Cd34 expression (either via violin or scatter plots) should be depicted to justify this approach.

– "To test this possibility, we isolated tibial periosteal mesenchymal progenitors…" Are you referring to Td+ cells from the intact bones of Col2-Cre mice?

– "These data clearly suggest that activin A targets mesenchymal progenitors in early fracture and that suppressing activin A impairs fracture healing." Activin A inhibition clearly disrupts fracture healing, but most of these individual results can also be explained by changes in chondrocytes, rather than PPCs, at the cellular level. This could perhaps be addressed by analyzing other modes of injury, such as cortical drill holes, that heal mainly through intramembranous ossification.

– "…is expressed highly in a previously unidentified PPC population emerging shortly after fracture…" This claim requires further support, as the uniqueness of the PPC population is unclear. These cells likely have at least some overlap with those described by others, such as Matthews et al. (i.e. alphaSMA) or Shi et al. (i.e. Gli1), and Col2-Cre could simply be another marker/handle for the same transient myofibroblast population during fracture healing.

– "…fibroblasts in all tissues, regardless of their states (steady and perturbed), are derived from a primitive cell cluster termed Pi16+ cells…" Buechler et al., identified two transcriptionally distinct populations of fibroblasts, that can be distinguished by Pi16 or Col15a1 expression, among other markers. Although both populations are found in various types of tissues, it is unclear whether the other tissue-specific fibroblasts are derived from them.

---

## [Author Response]

Essential revisions:This is a generally well-executed set of study, the data from which are appropriate and largely supports the conclusions. The manuscript would be strengthened by convincingly demonstrating that the PPCs are the primary mediator of Activin A actions during fracture repair.

We thank the Editors for their supportive and encouraging feedback. We also agree that our study does not provide direct evidence that PPCs are the primary mediator of Activin A actions during fracture repair. Thus, we have toned down our conclusions and we now include chondrocytes as an additional source of Activin A. Please see our responses to Reviewer 1 addressing this issue.

Reviewer #1 (Recommendations for the authors):Questions and concerns:1. Multiple pieces of data show Inhba/ACTIVIN A expression in a number of other cell types, but a majority of the language pins all of the responses to ACTIVIN A as coming from the PPCs and disregarding the other non-mesenchymal cell types. For example, in Sup. Figure 1 Inhba expression is quite high in the granulocyte clusters (13, 14, 15). While PPC produced ACTIVIN A is likely a strong participant, further knock-out data in the PPCs and other cell types of would be required to claim this. The final paragraph of the discussion includes this, but the language should be toned down throughout the results. If the language is to be kept, further cell specific KO data is required utilizing the Acta2 specific Cre KO of Inhba.

We fully agree with the Reviewer that genetic evidence would be needed to establish the roles of Activin A in different cell types including the PPCs. We recently received *Inhba* floxed mice from Dr. Martin Matzuk at Baylor University ^(1)^ and are in the process of validating the line. Because it would take a long time to perform the needed conditional gene knockout studies, we have followed the Reviewer’s advice and have toned down our language throughout the manuscript. Specifically, we include chondrocytes as another source of Activin A in the fracture callus and we re-wrote the first paragraph of the Discussion. We hope those changes address this concern fully.

2. Please ensure the proper formatting for genes/proteins is correct throughout the manuscript (ie when referring to the protein ACTIVIN A should be all caps).

Apologies for not making sure that genes/proteins are formatted correctly throughout the manuscript but we do so now. With regard to Activin A, however, we would respectfully ask to keep its format as is (i.e. Activin A). This is the way in which the protein is customarily identified, including in recent studies such as Latres et al., Nat Commun. 2017 Apr 28:15153 ^(2)^ and Ramachandran et al., EMBO J. 2021 Jul 15;40(14):e106317 ^(3)^. This would allow us to fit the literature in a seamless manner.

3. With the neutralizing antibody usage, how do you know it is specifically targeting the mesenchymal cells (page 13 top paragraph)? Only giving percentages of cells stained for pSMAD2 and aSMA does not explicitly demonstrate that these PPCs are the responsible party in decreasing fracture healing parameters. Co-staining as well as assessing potential decrease in pSMAD2 staining in other cell types may shed light on ACTIVIN A functions in other cell types.

We appreciate, and are thankful for, this comment. Given that the antibody is administered systemically, it is of course not possible to know if it specifically targeted the mesenchymal cells. As per Reviewer’s suggestion, we have calculated the percentage of pSMAD2+ cells within αSMA+ and αSMA- cell populations and found that after antibody administration, the percentage of pSMAD2+ cells within PPCs decreases but the percentage of pSMAD2+ cells within non-PPCs remains the same, suggesting that PPCs are a major target of antibody treatment. Thus, we have revised the corresponding portion of the Results as follows:

“To gain insights into whether the systemic nActA.AB administration affected the PPCs, we performed qRT-PCR and immunostaining analyses on early fracture samples from wild type mice as above. At day 7 post fracture, nActA.AB administration reduced the number of cells positive for phosphorylated SMAD2 (pSMAD2) through which Activin A normally signals intracellularly ^(4)^, suggesting the effectiveness of neutralizing antibody treatment (Figure 5E, F). Interestingly, the number of PPCs positive for αSMA and the percentage of pSMAD2+ cells within PPC population were significantly decreased, while the percentage of pSMAD2+ cells within non-PPCs remained the same. These data were further confirmed by reduced gene expression of *Acta2* and *Inhba* in fracture callus after nActA.AB administration (Figure 5G). Taken together, our results clearly suggest that the PPCs were the primary responsive cell type to Activin A in early fracture and that systemic interference of Activin A action by nActA.AB treatment impaired fracture healing.”

Furthermore, we similarly calculated the percentage of pSMAD2+ cells within αSMA+ and αSMA- cell populations in the callus after recombinant Activin A implantation and obtained the opposite results. Together, these complementary experiments strengthen our overall conclusions regarding the importance of Activin A in fracture repair.

4. What is the expression pattern of the ACTIVIN A receptor? This may allow for further conclusions. Additionally, in the in vitro MPC sorted rACTIVIN A experiments, markers of myofibroblast emerge after treatment and in the chondrogenic/osteogenic media there is further differentiation upon treatment. Are the authors suggesting there is a self-feedback mechanism on the PPCs from their own ACTIVIN A expression?

We greatly appreciate these comments and queries. We now include the expression patterns of Activin A receptors in Figure 3 —figure supplement 8C and include the following sentences in the Results:

“Activin A binds to type II receptors (ActRIIA or ActRIIB) to recruit and phosphorylate type I receptors (ALK4 or ALK7) for initiating its intracelluar signaling ^(4)^. UMAP plots suggested that genes encoding these receptors (*Acvr2a, Acvr2b, Acvr1b*, and *Acvr1c*, respectively) were expressed in all mesenchymal progenitor populations and *Acvr2a* expression was enriched in MPCs (Figure 3 —figure supplement 8C).”

These receptor expression patterns suggest that Activin A can act on most if not all mesenchymal subpopulations within the fracture callus, with a possible preference toward MPCs. Future experiments knocking out individual receptor in vitro and in vivo could further illustrate the exact cellular target(s) of Activin A during bone healing.

Regarding the interesting comment about a self-feedback mechanism, we do not have direct data sustaining or refuting such thesis. However, we describe in the first paragraph of Discussion a previous study on human mesenchymal stem cells in vitro showing that siRNA-mediated downregulation of endogenous *INHBA* expression resulted in inhibition of chondrogenesis and osteogenesis ^(5)^. The study provided clear evidence that the endogenous protein is needed for skeletogenic cell differentiation and may operate in a self-feedback manner. We hope to have described these results and their implications more clearly now.

5. Are cells in the bone only responsive to ACTIVIN A signaling after injury, or could there be effects of the protein in an uninjured setting?

We thank the Reviewer for this excellent suggestion. We performed µCT on the contralateral uninjured bones and found that administration of neutralizing antibody did not appreciably alter trabecular and cortical bone structure within the duration of our study. The data suggest that Activin A may not be essential for normal bone homeostasis, at least short-term. These data are now included as Figure 5 —figure supplement 10.

Reviewer #2 (Recommendations for the authors):In this study, Lutian Yao et al. have found that TGF-b family member Activin A is expressed in proliferative progenitor cells (PPCs) which was identified in the tibia by single cell analysis. They also demonstrated that PPCs gain myofibroblasts feature after fracture. They found that Activin A can stimulates proliferation and differentiation of periosteal progenitors and promote chondrogenesis and osteogenesis during fracture healing process. The authors have combined the fracture model with single cell analysis to investigate the function of Activin A during the bone healing process. Overall, the discovery from this study has important implications for the bone healing process during fracture. However, several important questions remain to be clarified in this manuscript. Moreover, the mechanistic insight of this manuscript still needs to be enhanced.1. In Figure 2D, the authors only showed the cell cycle stage for the populations they identified. The expression of cell proliferation marker also needs to be provided to further prove the proliferative feature of PPCs.

We fully agree and thank the Reviewer for raising this important issue. We now include violin plots of cell proliferation makers as Figure 2 —figure supplement 5 to address the Reviewer’s comment and strengthen our conclusions.

2. In Figure 2F and supplemental figure 6, the velocity analysis did not clearly show the trajectory of EOB. The authors should explain this.

This is a very good point. Based on the data in Figure 2F and Figure 2 —figure supplement 7, it appears quite clear that the MPCs serve as very early progenitors and give rise to PPCs, in turn developing into OBs (osteoblasts) and CHs (chondrocytes) over further time. RNA velocity also predicted that EOBs (early osteoblasts) not only develop into OBs as one would expect but may also contribute to the PPC population. Because these developmental trajectories are of course based on computational analysis, we feel we cannot speculate more as to their mechanistic basis and implications. We hope this explanation is sufficient for the Reviewer.

3. In Figure 3F, the staining of Activin A is confusing. Which color represents Activin A?

We apologize for the confusion. Activin A stain is in white. In the previous submission, we had identified the images by writing “Activin A” in black outside. For clarity, we now identify the images by writing “Activin A” in white inside, thus in the same color as the stain itself.

4. The authors have shown that Activin A can stimulate the proliferation and differentiation of periosteal progenitors in Figure 4. What is the mechanism behind this?

This is indeed a pertinent question. For the purpose of the present study, we did not directly test the cellular, biochemical and molecular mechanisms that could mediate and regulate Activin A action on the differentiation of periosteal progenitors. Based on literature, the binding of Activin A to one of its high affinity type II receptors (ActRIIA and ActRIIB) recruits and phosphorylates one of its low affinity type I receptors (ActRIs) -ALK4 and ALK7- with ALK4 being the predominant one. Once activated, ActRI phosphorylates Smad2 and 3 to form a heteromeric complex with Smad4, which transfers to the nucleus and interacts with cofactors to regulate the transcription of target genes ^(6)^. This canonical signaling pathway mediates major biological actions of Activin A on cell proliferation, differentiation, metabolism, repair, and apoptosis ^(7,8)^. Activated Smad complex also induces a negative feedback mechanism through the inhibitory Smad7 ^(9)^. Depending on cell types and physiological conditions, Activin A can also regulate noncanonical MAP kinases (p38, ERK, and *JNK*) ^(10-14)^. In addition, intrinsic antagonists for Activin A signaling have been identified, including Inhibin, follistatin (FST), FST-like 3 (FSTL3), Cripto, BAMBI etc. ^(15)^. It is our plan to identify the downstream pathways mediating Activin A action on periosteal progenitors in future studies.

5. The authors have demonstrated that Activin A can accelerate the fracture healing process. What is possible role of TGFb signaling in this process?

As the Reviewer correctly points out, the TGFβ pathway has long been known to regulate and promote fracture healing ^(16,17)^, with circulating levels of TGFβ1 and TGFβ2 as indicators of ongoing fracture repair ^(18)^. It has also been found that excessive levels of TGFβ can impair progenitor cell function and inhibit bone fracture healing ^(19)^. Thus, the TGFβ pathway has very important roles in bone healing but needs to be fine-tuned. It will be interesting to figure out in the future in what manner and to what extent TGFβ proteins and Activin A may interact to promote repair and whether they act on the exact same population(s) or stage(s) of the healing process.

Reviewer #3 (Recommendations for the authors):Below are some specific comments and questions for the authors:– It is unclear from Figure 1A if mesenchymal progenitors in bone marrow express Activin A. Based on the single cell RNA-seq data, it is possible that the majority of labeled marrow cells are granulocytes. This can be checked with higher magnification imaging of slides co-stained with CD31 (as the progenitors will likely be pericytes) or better yet LEPR.

Based on this suggestion, we performed Activin A and Emcn (a marker for endothelial cells) co-staining on mouse bone marrow. As shown in Author response image 1, the majority of Activin A+ bone marrow cells are not associated with vasculature, suggesting that they are not pericytes and may thus be granulocytes. It is well established in the literature that periosteal mesenchymal progenitors, but not bone marrow mesenchymal populations, play a major role in fracture healing ^(20)^. Thus, we focus our manuscript on periosteum before and after fracture.

**Author response image 1. sa2fig1:** Fluorescent Activin A (green) and Emcn (red) staining of bone marrow in a *WT* mouse femur.

– "Since almost all cells in the thickened periosteum at day 5 are Td+, this observation suggests that muscle-derived cells do not significantly contribute to the early stage of fracture healing". Not sure if your data supports this conclusion. Whatever portion of muscle cells participate in fracture healing, they may not express Col2-cre at baseline but start expressing it shortly after they are mobilized and differentiating, making it difficult to distinguish them from cells with periosteal or marrow origin in your system. Also, fibroadipogenic progenitors (if they are Td+) can be difficult to spot at the magnification the data are presented. Have you checked intact muscle tissue with FACS for Td+ cells?

We very much thank the Reviewer for pointing this out and we agree with the points raised. Thus, we removed that sentence in the revised manuscript. For Reviewer’s question, we did check muscle in uninjured legs. As shown in Author response image 2, we did not observe Td+ cells in the muscle from Col2/Td mice.

**Author response image 2. sa2fig2:** Fluorescent image of a *Col2/Td* tibia shows no Td signal in the neighboring muscle tissue.

– I am not sure if it is possible to call your cell collections exclusively "periosteal", given the anatomically broad recombination profile. The Methods sections suggests that the intact bones were not crushed, and it seems that the cell clusters do not include LEPR+/CAR cells, which are both reassuring. However, such control is difficult (or perhaps not even possible) when dealing with developing fracture callus.

We do very much agree that it is difficult to establish the exact nature and origin of the isolated cell populations. Because the dissected tibiae at day 0 and day 5 were sealed with agarose at each epiphyseal end prior to protease digestion and cell isolation, we are confident that marrow cells were not included in our samples. At the day 10 time points, we dissected out the callus mass microsurgically, avoiding the marrow area and surrounding tissues as much as possible. Note that the fracture was stabilized with intramedullary pins. Given the space the pin occupies, bone marrow area at the fracture region was much smaller than that in the intact bone. In addition, our previous study identified Adipoq-expressing marrow adipogenic lineage precursors (MALPs) in the bone marrow ^(21)^ but not at the periosteal surface (Author response image 3). Interestingly, we detected very few Adipoq+ cells in our fracture scRNA-seq datasets (Author response image 3), further confirming that cells we analyzed were largely if not exclusively periosteal, with few or no bone marrow cell contamination.

**Author response image 3. sa2fig3:** Adipoq+ cells are absent at mouse periosteum. (A) Immunofluorescent image of an intact tibiae from *Adipoq-Cre Td* mice stained with Endomucin (Emcn, an endothelial cell marker). (B) Expression pattern of *Adipoq* in mesenchymal lineage cells in the fracture scRNA-seq datasets.

– "Particularly relevant to the present study was the finding that compared to MPCs, PPCs highly expressed Inhba after fracture (Figure 3C and Supplementary Figure 7B). Note that chondrocytes also highly expressed Inhba, consistent with the immunostaining results shown in Figure 1. However, their number was much lower than PPCs in early callus (Supplementary Figure 4B), suggesting that PPCs are the main source of activin A in early fracture healing." The violin plot in Figure 3C alone does not support this statement. There appears to be a remarkable increase in Inhba expression in chondrocytes (which is consistent with Figure 1A-B), but less so in other cells. Further, SuppFig4B presents an estimate of cell counts, which might very well be biased by the efficiency of the flow cytometer and/or 10X platform, when comparing fibroblast-like cells and chondrocytes.

The Reviewer is absolutely correct. To address these very important points, we have rectified our description of the data and have modified our conclusions to include possible roles of the protein in diverse cell populations. One reason for our emphasizing the likely important roles of Inhba/Activin A in the progenitor cell populations is that as mentioned above, there is clear evidence of the protein’s role in MSC differentiation (for example, Djouad et al., 2010 cited above ^(5)^). To date however, Activin A roles in chondrocytes remain largely unclear, necessitating future studies. We agree also that there may be a bias in flow efficiency for different populations. Again, we have dealt with all these pertinent and important issues by toning down our conclusions and entertaining other possibilities. Please also see our response to Reviewer 1 comment 1.

– "Based on scRNA-seq data, we sorted Cd45-Cd31- Ter119-Cd34+ (Lin-/Cd34+) cells and Cd45-Cd31-Ter119-Cd34- (Lin-/Cd34-) cells to represent MPCs and PPCs, respectively." Cluster-specific quantification of Cd34 expression (either via violin or scatter plots) should be depicted to justify this approach.

Point well taken. The UMAP of Cd34 expression pattern is now included in Figure 2 —figure supplement 6B.

– "To test this possibility, we isolated tibial periosteal mesenchymal progenitors…" Are you referring to Td+ cells from the intact bones of Col2-Cre mice?

Sorry for the confusion. We actually used WT C57Bl/6 mice for these experiments. This detail is now included in Methods.

– "These data clearly suggest that activin A targets mesenchymal progenitors in early fracture and that suppressing activin A impairs fracture healing." Activin A inhibition clearly disrupts fracture healing, but most of these individual results can also be explained by changes in chondrocytes, rather than PPCs, at the cellular level. This could perhaps be addressed by analyzing other modes of injury, such as cortical drill holes, that heal mainly through intramembranous ossification.

We very much thank the Reviewer for this constructive advice and suggestions. To address this issue, we performed drill hole experiments on WT C56Bl/6 mice and treated them with subcutaneous injections of nActA.AB or implanted them with Matrigel/Activin A mixture at the hole site. MicroCT scanning clearly showed that bone healing in the hole region was delayed in the nActA.AB-treated group but accelerated in the recombinant Activin A-implanted group. As the Reviewer pointed out, bone repair after drill-hole is mainly through intramembranous ossification. Thus, these additional results provide evidence that compared with chondrocytes, PPCs could be more important in Activin A-mediated bone healing. These data are now included as Figure 7 in the revised manuscript.

– "…is expressed highly in a previously unidentified PPC population emerging shortly after fracture…" This claim requires further support, as the uniqueness of the PPC population is unclear. These cells likely have at least some overlap with those described by others, such as Matthews et al. (i.e. alphaSMA) or Shi et al. (i.e. Gli1), and Col2-Cre could simply be another marker/handle for the same transient myofibroblast population during fracture healing.

We fully agree with the Reviewer and have removed the “previously unidentified” terminology in the current revised manuscript. The Reviewer is also correct that the PPCs identified in our study are likely to be related to the αSMA+ cells discovered by the Matthews et al. study. In their *eLife* article ^(22)^, the authors concluded that αSMA identifies long-term, slow-cycling and self-renewing osteochondroprogenitors in adult periosteum as functionally important participants in bone formation and fracture healing. In our scRNA-seq datasets, however, *Acta2* mainly marks the PPCs after fracture and it is thus possible that the PPCs are a more specific subset amongst the populations identified by Matthews et al. Future experiments are needed to clarify these important points.

– "…fibroblasts in all tissues, regardless of their states (steady and perturbed), are derived from a primitive cell cluster termed Pi16+ cells…" Buechler et al., identified two transcriptionally distinct populations of fibroblasts, that can be distinguished by Pi16 or Col15a1 expression, among other markers. Although both populations are found in various types of tissues, it is unclear whether the other tissue-specific fibroblasts are derived from them.

We thank the Reviewer for this critical advice and for raising these points. As the Reviewer realizes, Buechler et al. stated the following in their article:

“In the steady state, slingshot lineage inference identified trajectories that emerged from the Pi16+ cluster, passed through the Col15a1+ cluster, and ended at specialized clusters.

In the perturbed state, universal Dpt+Pi16+ fibroblasts maintained the highest expression of stemness-associated genes (Extended Data Figure 8x). Lineage inference identified trajectories from Dpt+Pi16+ through Dpt+Col15a1+ and then on to perturbation-specific, activated Cxcl5+ and Lrrc15+ clusters or the Adamdec1+ cluster (Extended Data Figure 8y). We tested whether universal fibroblasts give rise to LRRC15+ myofibroblasts using a subcutaneous [tumor] model in the DptIresCreERT2;Rosa26LSLYFP mouse. We found that 52 ± 7% of LRRC15+ myofibroblasts were YFP+ in DptIresCreERT2ki/ki mice (Figure 3d, Extended Data Figure 8z–b′). This indicates that Dpt-expressing cells marked before [tumor] implantation can differentiate into LRRC15+ myofibroblasts”.

These authors’ statements clearly implicate the stemness of Pi16+ cells. However, we agree with the Reviewer that they did not provide sufficient evidence that all fibroblast populations derive from Pi16+ cells. Thus, we modified our own sentence as follows:

“Fibroblast populations present in all tissues regardless of physiologic state (steady and perturbed) contain a most primitive cell cluster termed Pi16+ cells”.

Interestingly, Col15a1 is expressed in both MPCs and PPCs in our datasets (Author response image 4), further demonstrating that fibroblasts (or mesenchymal lineage cells) in periosteum share characteristics of fibroblast populations present in other tissues.

**Author response image 4. sa2fig4:** UMAP plot of *Col15a1* in the merged fracture dataset.

References:

1. Pangas SA, Jorgez CJ, Tran M, Agno J, Li X, Brown CW, Kumar TR, Matzuk MM. Intraovarian activins are required for female fertility. Mol Endocrinol. 2007;21(10):2458-71.

2. Latres E, Mastaitis J, Fury W, Miloscio L, Trejos J, Pangilinan J, Okamoto H, Cavino K, Na E, Papatheodorou A, Willer T, Bai Y, Hae Kim J, Rafique A, Jaspers S, Stitt T, Murphy AJ, Yancopoulos GD, Gromada J. Activin A more prominently regulates muscle mass in primates than does GDF8. Nat Commun. 2017;8:15153.

3. Ramachandran A, Mehic M, Wasim L, Malinova D, Gori I, Blaszczyk BK, Carvalho DM, Shore EM, Jones C, Hyvonen M, Tolar P, Hill CS. Pathogenic ACVR1(R206H) activation by Activin A-induced receptor clustering and autophosphorylation. EMBO J. 2021;40(14):e106317.

4. Pangas SA, Woodruff TK. Activin signal transduction pathways. Trends Endocrinol Metab. 2000;11(8):309-14.

5. Djouad F, Jackson WM, Bobick BE, Janjanin S, Song Y, Huang GT, Tuan RS. Activin A expression regulates multipotency of mesenchymal progenitor cells. Stem Cell Res Ther. 2010;1(2):11.

6. Makanji Y, Zhu J, Mishra R, Holmquist C, Wong WP, Schwartz NB, Mayo KE, Woodruff TK. Inhibin at 90: from discovery to clinical application, a historical review. Endocr Rev. 2014;35(5):747-94.

7. Hedger MP, Winnall WR, Phillips DJ, de Kretser DM. The regulation and functions of activin and follistatin in inflammation and immunity. Vitam Horm. 2011;85:255-97.

8. Massague J. How cells read TGF-β signals. Nat Rev Mol Cell Biol. 2000;1(3):169-78.

9. Massague J, Seoane J, Wotton D. Smad transcription factors. Genes Dev. 2005;19(23):2783-810.

10. Bildik G, Akin N, Esmaeilian Y, Hela F, Yildiz CS, Iltumur E, Incir S, Karahuseyinoglu S, Yakin K, Oktem O. Terminal differentiation of human granulosa cells as luteinization is reversed by activin-A through silencing of *Jnk* pathway. Cell Death Discov. 2020;6(1):93.

11. Murase Y, Okahashi N, Koseki T, Itoh K, Udagawa N, Hashimoto O, Sugino H, Noguchi T, Nishihara T. Possible involvement of protein kinases and Smad2 signaling pathways on osteoclast differentiation enhanced by activin A. J Cell Physiol. 2001;188(2):236-42.

12. Hu J, Wang X, Wei SM, Tang YH, Zhou Q, Huang CX. Activin A stimulates the proliferation and differentiation of cardiac fibroblasts via the ERK1/2 and p38-MAPK pathways. Eur J Pharmacol. 2016;789:319-27.

13. de Guise C, Lacerte A, Rafiei S, Reynaud R, Roy M, Brue T, Lebrun JJ. Activin inhibits the human Pit-1 gene promoter through the p38 kinase pathway in a Smad-independent manner. Endocrinology. 2006;147(9):4351-62.

14. Bao YL, Tsuchida K, Liu B, Kurisaki A, Matsuzaki T, Sugino H. Synergistic activity of activin A and basic fibroblast growth factor on tyrosine hydroxylase expression through Smad3 and ERK1/ERK2 MAPK signaling pathways. J Endocrinol. 2005;184(3):493-504.

15. Harrison CA, Gray PC, Vale WW, Robertson DM. Antagonists of activin signaling: mechanisms and potential biological applications. Trends Endocrinol Metab. 2005;16(2):73-8.

16. Borton AJ, Frederick JP, Datto MB, Wang XF, Weinstein RS. The loss of Smad3 results in a lower rate of bone formation and osteopenia through dysregulation of osteoblast differentiation and apoptosis. J Bone Miner Res. 2001;16(10):1754-64.

17. Blumenfeld I, Srouji S, Lanir Y, Laufer D, Livne E. Enhancement of bone defect healing in old rats by TGF-β and IGF-1. Exp Gerontol. 2002;37(4):553-65.

18. Chaverri D, Vivas D, Gallardo-Villares S, Granell-Escobar F, Pinto JA, Vives J. A pilot study of circulating levels of TGFbeta1 and TGF-beta2 as biomarkers of bone healing in patients with non-hypertrophic pseudoarthrosis of long bones. Bone Rep. 2022;16(doi):101157.

19. Liu J, Zhang J, Lin X, Boyce BF, Zhang H, Xing L. Age-associated callus senescent cells produce TGF-Î²1 that inhibits fracture healing in aged mice. J Clin Invest. 2022;132(8):e148073.

20. Bragdon BC, Bahney CS. Origin of Reparative Stem Cells in Fracture Healing. Curr Osteoporos Rep. 2018;16(4):490-503.

21. Zhong L, Yao L, Tower RJ, Wei Y, Miao Z, Park J, Shrestha R, Wang L, Yu W, Holdreith N, Huang X, Zhang Y, Tong W, Gong Y, Ahn J, Susztak K, Dyment N, Li M, Long F, Chen C, Seale P, Qin L. Single cell transcriptomics identifies a unique adipose lineage cell population that regulates bone marrow environment. *eLife*. 2020;9:e54695.

22. Matthews BG, Novak S, Sbrana FV, Funnell JL, Cao Y, Buckels EJ, Grcevic D, Kalajzic I. Heterogeneity of murine periosteum progenitors involved in fracture healing. *eLife*. 2021;10:e58534.